# HIGH-ORDER TENSOR RECOVERY WITH A TENSOR $U_1$ NORM

## ABSTRACT

Recently, numerous tensor SVD (t-SVD)-based tensor recovery methods have emerged, showing promise in processing visual data. However, these methods often suffer from performance degradation when confronted with high-order tensor data exhibiting non-smooth changes, commonly observed in real-world scenarios but ignored by the traditional t-SVD-based methods. Our objective in this study is to provide an effective tensor recovery technique for handling non-smooth changes in tensor data and efficiently exploring the correlations of high-order tensor data across its various dimensions, without the need for introducing numerous variables and weights. To this end, we introduce a new tensor decomposition and a new tensor norm called the Tensor $U_1$ norm. **We utilize these novel techniques in solving the problem of high-order tensor completion problem and provide theoretical guarantees for the exact recovery of the resulting tensor completion models.** An optimization algorithm is proposed to solve the resulting tensor completion model iteratively by combining the proximal algorithm with the Alternating Direction Method of Multipliers. Theoretical analysis showed the convergence of the algorithm to the Karush–Kuhn–Tucker (KKT) point of the optimization problem. Numerical experiments demonstrated the effectiveness of the proposed method in high-order tensor completion, especially for tensor data with non-smooth changes.

## 1 INTRODUCTION

In recent years, lots of tensor methods have been proposed for better studying the low-rankness in massive high-dimensional tensor data such as color images, hyperspectral images, and videos Kajo et al. (2019); Lu et al. (2019a); Zhang et al. (2014); Lu et al. (2019b; 2018); Madathil & George (2018), as the traditional matrix methods Candes & Plan (2010); Candès & Recht (2009); Candès et al. (2011); Chandrasekaran et al. (2009); Xu et al. (2012); Wright et al. (2009) fail on handling the tensor data. Depending on different adopted low-rank prior, these tensor methods can be mainly categorized as: (1) CP (Canonical Polyadic) Decomposition-based methods Hitchcock (1927; 1928); Kolda & Bader (2009), (2) Tucker Decomposition-based methods Tucker (1963); Gandy et al. (2011); Liu et al. (2013), and (3) t-SVD-based methods Lu et al. (2019b; 2018); Zheng et al. (2020); Qin et al. (2022).

Recently, the t-SVD-based methods for three-order tensor recovery have garnered growing attention and achieved great success in the applications of visual data processing, such as data denoising Lu et al. (2019a), image and video inpainting Zhang et al. (2014); Lu et al. (2019b; 2018), and background modeling and initialization Lu et al. (2019a); Kajo et al. (2018). In these t-SVD-based methods, a fixed invertible transform, such as Discrete Fourier Transform (DFT) Lu et al. (2018) and Discrete Cosine Transform (DCT) Lu et al. (2019b), is applied to a three-order tensor data along its third dimension, and the low-rankness prior of each frontal slice of the resulting tensor (corresponding to the different frequency information along the third dimension of tensor data) has been used to explore global low-rankness of tensor data.

However, the approach of performing invertible transform and analyzing the low rankness of the slices along a specific dimension of tensor data poses challenges in generalizing t-SVD-based methods directly to higher order tensor cases, such as color video, compared to CP decomposition-based and Tucker Decomposition-based methods. A common solution for easily handling the high-order tensors is to utilize tensor unfolding operators Zheng et al. (2020); Qin et al. (2022). For example, in Zheng et al. (2020), the Weighted Sum of Tensor Nuclear Norm of all mode-$k_1 k_2$ unfolding

Table 1: Illustration to the slice permutation variability (SPV) of t-SVD: mean PSNR (MPSNR) results for same tensor data (including *templet*, *bus*, and *mobile*.) but with different slice order for video reconstruction (best rank 50 estimating) are reported.

| Random shuffling for frames | DFT-based t-SVD | | | DCT-based t-SVD | | |
|:---:|:---:|:---:|:---:|:---:|:---:|:---:|
| | mobile | bus | tempete | mobile | bus | tempete |
| No | 25.74 | 30.22 | 29.43 | 24.81 | 29.74 | 29.00 |
| Yes | 23.80 | 28.71 | 28.58 | 23.79 | 28.72 | 28.54 |

tensors (WSTNN) has been proposed to study the correlations along different modes of higher order tensors. It is worth noting that $\binom{h}{2}$ different unfolding tensors are considered in WSTNN for $h$-order tensor recovery, which leads to a difficult setting for tunable parameters. On the other hand, as shown in Table 1[1], the t-SVD-based methods exhibit tensor slices permutation variability (SPV). **That is, interchanging the frontal slice of the tensor affects the effectiveness of tensor recovery significantly.** This is simply because a fixed invertible transform, such as DFT or DCT, is performed on tensor along certain dimensions, **making the t-SVD sensitive to non-smooth changes in tensor slices, particularly when the sequence of these slices is disordered. This scenario is frequently observed within classification tasks, as the sequence of samples is often disordered prior to classification.** Although Zheng et al. (2022) have proposed an effective solution by solving a Minimum Hamiltonian circle problem for the case of DFT, a general solution is still lacking as the similarity among slices does not consistently correlate with the tensor's low-rank property, in contexts beyond DFT. For the same reason (the use of a fixed transform), the recovery performance of the t-SVD-based methods is also susceptible to non-continue and non-smooth of the tensor data itself Kong et al. (2021). These data include videos with rapidly changing content between frames or tensor data obtained by concatenating different scenes.

This study aims to address these challenges by developing an efficient tensor recovery technique. To the best of our knowledge, this is the first work to investigate the impact of non-smooth changes on t-SVD-based methods for high-order tensor recovery and provide an effective solution. Our contributions can be summarized as follows:

- Based on the t-SVD-based methods, we proposed a new tensor decomposition named Tensor Decomposition Based on Slices-Wise Low-Rank Prior (TDSL) by introducing a set of unitary matrices that can be either learned (to effectively address SPV and non-smoothness issues in the traditional t-SVD-based methods) or derived from known prior information, thereby allowing our model to harness the inherent data characteristics.

- To address challenges in generalizing t-SVD-based methods directly to higher-order tensor cases, we give a new tensor norm named the Tensor $U_1$ norm, which is designed for extending TDSL further to higher-order tensor completion without the need for additional variables and weights.

- To solve the resulting models (TC-SL and TC-U1) based on TDSL and Tensor $U_1$ norm effectively, we present an optimization algorithm that combines the proximal algorithm with the Alternating Direction Method of Multipliers, along with a corresponding convergence analysis in theory. **We theoretical analysis the conditions for exact recovery of the proposed models, and give upper bound of estimation errors by given methods when tensor data contaminated by noise.** Our experiments on synthetic data demonstrate the exact recovery of TC-U1 under fairly broad conditions.

## 2 TENSOR RECOVERY WITH NEW TENSOR NORM

Before introducing the proposed methods, we summarize some notations in Table 2 that will be used later.

---

[1]The three gray videos are chosen from `http://trace.eas.asu.edu/yuv/`

Table 2: Notations

| Notations | Descriptions | Notations | Descriptions |
|---|---|---|---|
| $a, b, c, \cdots$ | scalars | $\boldsymbol{a}, \boldsymbol{b}, \boldsymbol{c}, \cdots$ | vectors |
| $\boldsymbol{A}, \boldsymbol{B}, \boldsymbol{C}, \cdots$ | matrices | $\boldsymbol{\mathcal{A}}, \boldsymbol{\mathcal{B}}, \boldsymbol{\mathcal{C}}, \cdots$ | tensors |
| $\mathbb{A}, \mathbb{B}, \mathbb{C}, \cdots$ | sets | $\mathcal{A}, \mathcal{B}, \mathcal{C}, \cdots$ | operators |
| $\boldsymbol{\mathcal{I}}$ | identity tensor | $\mathbf{0}$ | null tensor |
| $\boldsymbol{A}^T$ | conjugate transpose of $\boldsymbol{A}$ | $\boldsymbol{F}_n$ | **Discrete Fourier Matrix (DFM)** with size of $n \times n$ |
| $[\boldsymbol{\mathcal{A}}]_{i_1, i_2, \cdots, i_h}$ | $(i_1, i_2, \cdots, i_h)$-th element of $\boldsymbol{\mathcal{A}}$ | $\boldsymbol{\mathcal{A}}_{(k)}$ (or $[\boldsymbol{\mathcal{A}}]_{(n)}$) | Mode-n Unfolding of $\boldsymbol{\mathcal{A}}$ |
| $\mathbb{A}^c$ | complementary set of $\mathbb{A}$ | $|\mathbb{A}|$ | number of elements of $\mathbb{A}$ |
| $[\boldsymbol{\mathcal{A}}]_{:,:,i_3,\cdots,i_h}$ | slice along the $(k_1, k_2)$-th mode | $\times_n$ | Mode-n product |
| $\|\boldsymbol{\mathcal{A}}\|_0$ | the number of non-zero elements of $\boldsymbol{\mathcal{A}}$ | $\|\boldsymbol{\mathcal{A}}\|_1$ | $\|\boldsymbol{\mathcal{A}}\|_1 = \sum_{i_1,i_2,\cdots,i_h} |[\boldsymbol{\mathcal{A}}]_{i_1,i_2,\cdots,i_h}|$ |

## 2.1 TENSOR DECOMPOSITION BASED ON SLICES-WISE LOW-RANK PRIORS (TDSL)

Differing from CP rank and Tucker rank, the t-SVD-based tensor rank operates on the assumption that an $h$-order tensor **data** $\boldsymbol{\mathcal{A}} \in \mathbb{R}^{I_1 \times I_2 \times \cdots \times I_h}$ **in** the real world can be decomposed as

$$\boldsymbol{\mathcal{A}} = \boldsymbol{\mathcal{Z}} \times_{k_3} \boldsymbol{L}_{k_3}^T \cdots \times_{k_h} \boldsymbol{L}_{k_h}^T, \tag{1}$$

where $\{\boldsymbol{L}_{k_t}\}_{t=3}^h$ are a set of given invertible transforms, and $\text{rank}_{(k_1,k_2)}(\boldsymbol{\mathcal{Z}}) << \min(I_{k_1}, I_{k_2})$. Here, $\text{rank}_{(k_1,k_2)}(\boldsymbol{\mathcal{Z}}) = \max_{i_{k_3}, i_{k_4}, \cdots, i_{k_h}} \text{rank}\left([\boldsymbol{\mathcal{Z}}]_{:,:,i_{k_3},\cdots,i_{k_h}}\right)$ is called as the slice wise tensor rank along the $(k_1, k_2)$-th mode of $\boldsymbol{\mathcal{Z}}$. If taken $\{\boldsymbol{L}_{k_t}\}_{t=3}^h$ as $\{\boldsymbol{F}_{I_{k_t}}\}_{t=3}^h$, $\text{rank}_{(k_1,k_2)}(\boldsymbol{\mathcal{Z}})$ is referred as to **tensor tubal rank** of $\boldsymbol{\mathcal{A}}$ Lu et al. (2018). And we can observe that when an $h$-order tensor data $\boldsymbol{\mathcal{A}}$ exhibits smooth changes along the $k$-th dimension, the mode-$k$ unfolding matrix of $\boldsymbol{\mathcal{Z}}$ is approximately column-sparse, meaning that most columns of the unfolding matrix are close to $\mathbf{0}$. Therefore, $\boldsymbol{\mathcal{A}}$ will be approximately **low tensor tubal rank** if it shows smoothness across all dimensions. However, this assumption is rarely met in real-world scenarios. For instance, tensor data may exhibit non-smooth changes caused by disordered images sequence, which often occur in image classification tasks Zheng et al. (2022). (Similar behavior can be observed for other invertible transforms, such as DCT.) In such cases, there exists a permutation matrix $\boldsymbol{P}$ such that $\boldsymbol{\mathcal{A}} \times_{k_n} \boldsymbol{P} = \boldsymbol{\mathcal{Z}} \times_{k_3} \boldsymbol{L}_{k_3}^T \cdots \times_{k_n} (\boldsymbol{L}_{k_n}\boldsymbol{P})^T \times k_n \cdots \times_{k_h} \boldsymbol{L}_{k_h}^T$ exhibits a better low-rank property than $\boldsymbol{\mathcal{A}}$. Furthermore, to deal with other cases where the initially given transform doesn't work, such as videos with rapidly changing content, it becomes necessary to learn more suitable invertible transforms. Therefore, we introduce a set of learnable unitary matrices to (1).

Additionally, leveraging fixed transforms based on known priors for tensor data can lead to effective tensor decomposition and reduce the computational burden associated with updating the learnable unitary matrices. Therefore, we incorporate fixed transforms $\{\hat{\boldsymbol{U}}_{k_n}\}_{n=3}^s$ determined by the priors of the tensor data. For instance, in the case of a color image with smoothness along its first dimension, we can set $\hat{\boldsymbol{U}}_1$ as DFM or **Discrete Cosine Matrix (DCM)**. Given the fixed transforms $\{\hat{\boldsymbol{U}}_{k_n}\}_{n=3}^s$, we can find a value $0 \leq r \leq \min(I_{k_1}, I_{k_2})$ such that

$$\boldsymbol{\mathcal{A}} = \boldsymbol{\mathcal{Z}}_1 \times_{k_3} \hat{\boldsymbol{U}}_{k_3}^T \cdots \times_{k_s} \hat{\boldsymbol{U}}_{k_s}^T \times_{k_{s+1}} \boldsymbol{U}_{k_{s+1}}^T \cdots \times_{k_h} \boldsymbol{U}_{k_h}^T, \tag{2}$$

where $\text{rank}_{(k_1,k_2)}(\boldsymbol{\mathcal{Z}}_1) = r$, and $\{\boldsymbol{U}_{k_n}\}_{n=s+1}^h$ are the unitary matrices. For any given value of $r$ and tensor $\boldsymbol{\mathcal{A}}$, we can obtain an approximate tensor decomposition for $\boldsymbol{\mathcal{A}}$ using Algorithm 2 given in the supplementary material. We refer to $\boldsymbol{\mathcal{A}}$ as having slices-wise low-rankness along the $(k_1, k_2)$-the mode if $\text{rank}_{(k_1,k_2)}(\boldsymbol{\mathcal{Z}}_1) << \min(I_{k_1}, I_{k_2})$. We denote (2) as Tensor Decomposition Based on Slices-Wise Low-Rank Priors (TDSL) of $\boldsymbol{\mathcal{A}}$.

Considering the tensor data may with missing elements, we apply the proposed TDSL to Tensor Completion and obtain the following model (TC-SL) based on the given $\{\hat{\boldsymbol{U}}_{k_n}\}_{n=3}^s$:

$$\min_{\boldsymbol{\mathcal{X}}, \boldsymbol{\mathcal{Z}}, \boldsymbol{U}_{k_n}^T \boldsymbol{U}_{k_n} = \boldsymbol{I}(n=s+1,\cdots,h)} \|\boldsymbol{\mathcal{Z}}\|_{*,\mathcal{U}}^{(k_1,k_2)}$$

$$s.t.\ \Psi_{\mathbb{I}}(\boldsymbol{\mathcal{M}}) = \Psi_{\mathbb{I}}(\boldsymbol{\mathcal{X}}),\ \boldsymbol{\mathcal{X}} = \boldsymbol{\mathcal{Z}} \times_{k_{s+1}} \boldsymbol{U}_{k_{s+1}}^T \cdots \times_{k_h} \boldsymbol{U}_{k_h}^T, \tag{3}$$

where $\|\mathcal{Z}\|_{*,\mathcal{U}}^{(k_1,k_2)} = \sum_{i_{k_3},i_{k_4},\cdots,i_{k_h}} \|[\mathcal{U}(\mathcal{Z})]_{:,:,i_{k_3},i_{k_4},\cdots,i_{k_h}}\|_*, \mathcal{U}(\mathcal{Z}) = \mathcal{Z} \times_{k_3} \hat{U}_{k_3} \cdots \times_{k_s} \hat{U}_{k_s}$, and $\Psi_{\mathbb{I}}$ is a linear project operator on the support set $\mathbb{I}$ such that

$$[\Psi_{\mathbb{I}}(\mathcal{M})]_{i_1,i_2,\cdots,i_h} = \begin{cases} [\mathcal{M}]_{i_1,i_2,\cdots,i_h}, & \text{if } (i_1, i_2, \cdots, i_h) \in \mathbb{I}; \\ 0, & \text{if } (i_1, i_2, \cdots, i_h) \notin \mathbb{I}. \end{cases} \tag{4}$$

**From the result given in Lu et al. (2019b), we know that, when $h = 3$ and $U_3$ is orthogonal, $\|\cdot\|_{*,\mathcal{U}}^{(1,2)}$ is the tightest convex envelope of $\mathrm{rank}_{(1,2)}(\mathcal{U}(\cdot))$ on the set $\{\mathcal{A}|\|\mathcal{A}\|_{2,\mathcal{U}}^{(1,2)} \leq 1\}$, where $\|\mathcal{A}\|_{2,\mathcal{U}}^{(1,2)} = \max_{i_3} \|[\mathcal{U}(\mathcal{A})]_{:,:,i_3}\|_2$. This conclusion can be easily extended to our case, and thus we have the following conclusion.**

**Property 1.** $\|\cdot\|_{*,\mathcal{U}}^{(k_1,k_2)}$ *is the dual norm of tensor $\|\cdot\|_{2,\mathcal{U}}^{(k_1,k_2)}$ norm, and $\|\cdot\|_{*,\mathcal{U}}^{(k_1,k_2)}$ is the tightest convex envelope of $\mathrm{rank}_{(k_1,k_2)}(\mathcal{U}(\cdot))$ on the set $\{\mathcal{A}|\|\mathcal{A}\|_{2,\mathcal{U}}^{(k_1,k_2)} \leq 1\}$.*

## 2.2 Tensor Completion with Tensor $U_1$ Norm

From (2), we observe that the Tensor Decomposition Based on Slices-Wise Low-Rank Priors (TDSL) of $\mathcal{A}$ depends on the choice of $k_1$ and $k_2$, and it considers different kinds of low-rankness of information in the tensor data for different $(k_1, k_2)$. Therefore, if we only consider one mode, it may lead to the loss of correlation information across the remaining modes. **Inspired by the sparsity prior in natural signals Wright et al. (2008), *i.e.*, the natural signals are often sparse if expressed on a proper basis, we propose an alternative approach by assuming that $\mathcal{Z}_1$ exhibits sparsity along its $(k_1, k_2)$-th mode, *i.e.*, there exists $U_{k_1}$ and $U_{k_2}$ such that $\mathcal{Z}_2 = \mathcal{Z}_1 \times_{k_1} U_{k_1} \times_{k_2} U_{k_2}$ is sparse.** Therefore, we present the following tensor decomposition:

$$\mathcal{A} = \mathcal{Z}_2 \times_{k_1} U_{k_1}^T \times_{k_2} U_{k_2}^T \times_{k_3} \hat{U}_{k_3}^T \cdots \times_{k_s} \hat{U}_{k_s}^T \times_{k_{s+1}} U_{k_{s+1}}^T \cdots \times_{k_h} U_{k_h}^T. \tag{5}$$

We refer to (5) as Tensor Decomposition Based on Sparsity Priors (TDST) for given fixed transforms $\{\hat{U}_{k_n}\}_{n=3}^s$ of $\mathcal{A}$. Comparing (5) with the CP decomposition and Tucker decomposition, we find that if a tensor has a low CP rank or Tucker rank, there exists $\{U_k\}_{k=1}^h$ such that $\mathcal{A} \times_1 U_1 \times_2 U_2 \cdots \times_h U_h = \mathcal{Z}_2$ being a sparse tensor.

Based on TDST, we get the following tensor completion model by using given $\{\hat{U}_{k_n}\}_{n=1}^s$:

$$\min_{\mathcal{X},\mathcal{Z},U_{k_n}^T U_{k_n}=I(n=s+1,\cdots,h)} \|\mathcal{Z}\|_{\mathcal{U},0} \tag{6}$$

$$s.t. \ \Psi_{\mathbb{I}}(\mathcal{M}) = \Psi_{\mathbb{I}}(\mathcal{X}), \ \mathcal{X} = \mathcal{Z} \times_{k_{s+1}} U_{k_{s+1}}^T \cdots \times_{k_h} U_{k_h}^T, \tag{7}$$

where $\|\mathcal{Z}\|_{\mathcal{U},0} = \|\mathcal{U}(\mathcal{Z})\|_0$ named tensor $U_0$ norm of $\mathcal{Z}$, and $\mathcal{U}(\mathcal{Z}) = \mathcal{Z} \times_{k_1} \hat{U}_{k_1} \cdots \times_{k_s} \hat{U}_{k_s}$. Since the tensor $U_0$ norm is discrete and non-convex, it will make related optimization problems NP-hard. Therefore, we introduce the following convex tensor norm.

**Definition 1.** *(Tensor $U_1$ Norm) For any tensor $\mathcal{A} \in \mathbb{R}^{I_1 \times I_2 \times \cdots \times I_h}$ and given $\{\hat{U}_k\}_{k=1}^h$, the tensor $U_1$ norm of $\mathcal{A}$ is defined as $\|\mathcal{A}\|_{\mathcal{U},1} = \|\mathcal{U}(\mathcal{A})\|_1$, where $\mathcal{U}(\mathcal{A}) = \mathcal{A} \times_1 \hat{U}_1 \times_2 \cdots \times_h \hat{U}_h$.*

As we will see in Property 2, the tensor $U_1$ norm is the tightest convex envelope of the tensor $U_0$ norm within the unit sphere of the tensor $U_\infty$ norm given in Definition 2. The proof of the Property 2 is given in the supplementary material.

**Definition 2.** *(Tensor $U_\infty$ Norm) For any tensor $\mathcal{A} \in \mathbb{R}^{I_1 \times I_2 \times \cdots \times I_h}$, the tensor $U_1$ norm of $\mathcal{A}$ is defined as $\|\mathcal{A}\|_{\mathcal{U},\infty} = \|\mathcal{U}(\mathcal{A})\|_\infty$, where $\mathcal{U}(\mathcal{A}) = \mathcal{A} \times_1 \hat{U}_1 \times_2 \cdots \times_h \hat{U}_h$.*

**Property 2.** *Given $\{\hat{U}_k\}_{k=1}^h$, we have the following conclusions for the Definitions 1-2:*

   (i) **Convexity:** *tensor $U_1$ norm and tensor $U_\infty$ norm are convex;*

   (ii) **Duality:** *tensor $U_1$ norm is the dual norm of tensor $U_\infty$ norm;*

   (iii) **Convex Envelope:** *tensor $U_1$ norm is the tightest convex envelope of the tensor $U_0$ norm within $\mathbb{S} = \{\mathcal{A}|\|\mathcal{A}\|_{\mathcal{U},\infty} \leq 1\}$.*

Therefore, based on the tensor $U_1$ norm, we get the following model (TC-U1):

$$\min_{\mathcal{X},\mathcal{Z},U_{k_n}^T U_{k_n}=I(n=s+1,\cdots,h)} \|\mathcal{Z}\|_{\mathcal{U},1}$$

$$s.t. \ \Psi_{\mathbb{I}}(\mathcal{M}) = \Psi_{\mathbb{I}}(\mathcal{X}), \ \mathcal{X} = \mathcal{Z} \times_{k_{s+1}} U_{k_{s+1}}^T \cdots \times_{k_h} U_{k_h}^T. \tag{8}$$

# 3 OPTIMIZATION ALGORITHM

In this section, we will provide a detailed discussion of the optimization of (8). Due to space limitations, we will leave out the optimization of (3), as we can obtain a similar result by following the same idea.

## 3.1 OPTIMIZATION ALGORITHM FOR TC-U1

For easily solving the problem (8), we introduce auxiliary variable $\boldsymbol{\mathcal{E}} \in \mathbb{E} = \{\boldsymbol{\mathcal{E}} | \Psi_{\mathbb{I}}(\boldsymbol{\mathcal{E}}) = \boldsymbol{0}\}$. Therefore, we turn to solve the following equivalence problem:

$$\min_{\boldsymbol{\mathcal{Z}}, \boldsymbol{U}_{k_n}^T \boldsymbol{U}_{k_n} = \boldsymbol{I}(n = s+1, \cdots, h)} \|\boldsymbol{\mathcal{Z}}\|_{\mathcal{U},1} \qquad s.t. \ \Psi_{\mathbb{I}}(\boldsymbol{\mathcal{M}}) = \boldsymbol{\mathcal{Z}} \times_{k_{s+1}} \boldsymbol{U}_{k_{s+1}}^T \cdots \times_{k_h} \boldsymbol{U}_{k_h}^T + \boldsymbol{\mathcal{E}}. \quad (9)$$

The Lagrangian function of (9) is formulated as

$$\mathcal{L}(\boldsymbol{\mathcal{Z}}, \{\boldsymbol{U}_{k_n}\}_{n=s+1}^h, \boldsymbol{\mathcal{E}}, \boldsymbol{\mathcal{Y}}, \mu) = \|\boldsymbol{\mathcal{Z}}\|_{\mathcal{U},1} + \langle \Psi_{\mathbb{I}}(\boldsymbol{\mathcal{M}}) - \boldsymbol{\mathcal{Z}} \times_{k_{s+1}} \boldsymbol{U}_{k_{s+1}}^T \cdots \times_{k_h} \boldsymbol{U}_{k_h}^T - \boldsymbol{\mathcal{E}}, \boldsymbol{\mathcal{Y}} \rangle$$

$$+ \frac{\mu}{2} \|\Psi_{\mathbb{I}}(\boldsymbol{\mathcal{M}}) - \boldsymbol{\mathcal{Z}} \times_{k_{s+1}} \boldsymbol{U}_{k_{s+1}}^T \cdots \times_{k_h} \boldsymbol{U}_{k_h}^T - \boldsymbol{\mathcal{E}}\|_F^2, \quad (10)$$

where $\boldsymbol{\mathcal{E}} \in \mathbb{E}$, $\boldsymbol{\mathcal{Y}}$ is Lagrange multiplier, and $\mu$ is a positive scalar. We solve (9) iteratively by combining the proximal algorithm with the Alternating Direction Method of Multipliers (PADMM) as follows.

**Step 1** Calculate $\boldsymbol{\mathcal{Z}}^{(t+1)}$ by

$$\boldsymbol{\mathcal{Z}}^{(t+1)} = \arg\min_{\boldsymbol{\mathcal{Z}}} \|\boldsymbol{\mathcal{Z}}\|_{\mathcal{U},1} + \frac{\mu^{(t)}}{2} \|\hat{\boldsymbol{\mathcal{P}}} \times_{k_{s+1}} \boldsymbol{U}_{k_{s+1}}^{(t)} \cdots \times_{k_h} \boldsymbol{U}_{k_h}^{(t)} - \boldsymbol{\mathcal{Z}}\|_F^2 + \frac{\eta^{(t)}}{2} \|\boldsymbol{\mathcal{Z}}^{(t)} - \boldsymbol{\mathcal{Z}}\|_F^2$$

$$= \mathcal{U}^{-1} \Big( \Big( \mathcal{U} \Big( \frac{\mu^{(t)} \hat{\boldsymbol{\mathcal{P}}} \times_{k_{s+1}} \boldsymbol{U}_{k_{s+1}}^{(t)} \cdots \times_{k_h} \boldsymbol{U}_{k_h}^{(t)} + \eta^{(t)} \boldsymbol{\mathcal{Z}}^{(t)}}{\mu^{(t)} + \eta^{(t)}} \Big), \frac{1}{\mu^{(t)} + \eta^{(t)}} \Big)_+ \Big), \quad (11)$$

where $\hat{\boldsymbol{\mathcal{P}}} = \Psi_{\mathbb{I}}(\boldsymbol{\mathcal{M}}) - \boldsymbol{\mathcal{E}}^{(t)} + \frac{1}{\mu^{(t)}} \boldsymbol{\mathcal{Y}}^{(t)}$.

**Step 2** Calculate $\boldsymbol{U}_{k_n}^{(t+1)}$ $(s+1 \leq n \leq h)$ by

$$\boldsymbol{U}_{k_n}^{(t+1)} = \arg\min_{\boldsymbol{U}_{k_n}^T \boldsymbol{U}_{k_n} = \boldsymbol{I}} \frac{\mu^{(t)}}{2} \|\hat{\boldsymbol{\mathcal{P}}} \times_{k_{s+1}} \boldsymbol{U}_{k_{s+1}}^{(t+1)} \cdots \times_{k_n} \boldsymbol{U}_{k_n} - \boldsymbol{\mathcal{Z}}^{(t+1)} \times_{k_h} \boldsymbol{U}_{k_h}^{(t)T}$$

$$\times_{k_h-1} \cdots \times_{k_n+1} \boldsymbol{U}_{k_n+1}^{(t)T}\|_F^2 + \frac{\eta^{(t)}}{2} \|\boldsymbol{U}_{k_n}^{(t)} - \boldsymbol{U}_{k_n}\|_F^2. \quad (12)$$

Since

$$\mu^{(t)} \|\hat{\boldsymbol{\mathcal{P}}} \times_{k_{s+1}} \boldsymbol{U}_{k_{s+1}}^{(t+1)} \cdots \times_{k_n} \boldsymbol{U}_{k_n} - \boldsymbol{\mathcal{Z}}^{(t+1)} \times_{k_h} \boldsymbol{U}_{k_h}^{(t)T} \cdots \times_{k_n+1} \boldsymbol{U}_{k_n+1}^{(t)T}\|_F^2 + \eta^{(t)} \|\boldsymbol{U}_{k_n} - \boldsymbol{U}_{k_n}^{(t)}\|_F^2$$

$$= \mu^{(t)} \|\boldsymbol{U}_{k_n} \boldsymbol{\mathcal{B}}_{(k_n)} - \boldsymbol{\mathcal{A}}_{(k_n)}\|_F^2 + \eta^{(t)} \|\boldsymbol{U}_{k_n} - \boldsymbol{U}_{k_n}^{(t)}\|_F^2$$

$$= \|\boldsymbol{U}_{k_n} [\sqrt{\mu^{(t)}} \boldsymbol{\mathcal{B}}_{(k_n)}, \sqrt{\eta^{(t)}} \boldsymbol{I}] - [\sqrt{\mu^{(t)}} \boldsymbol{\mathcal{A}}_{(k_n)}, \sqrt{\eta^{(t)}} \boldsymbol{U}_{k_n}^{(t)}]\|_F^2, \quad (13)$$

the optimal solution of (12) can be given by $\boldsymbol{U}_{k_n}^{(t+1)} = \boldsymbol{U}\boldsymbol{V}^T$ Zou et al. (2006), where $\boldsymbol{\mathcal{A}} = \boldsymbol{\mathcal{Z}}^{(t+1)} \times_{k_h} \boldsymbol{U}_{k_h}^{(t)T} \times_{k_h-1} \cdots \times_{k_n+1} \boldsymbol{U}_{k_n+1}^{(t)T}$, $\boldsymbol{\mathcal{B}} = \hat{\boldsymbol{\mathcal{P}}} \times_{s+1} \boldsymbol{U}_{k_{s+1}}^{(t+1)} \cdots \times_{k_n-1} \boldsymbol{U}_{k_n-1}^{(t+1)}$, $\boldsymbol{U}$ and $\boldsymbol{V}$ can be obtained by SVD of $[\sqrt{\mu^{(t)}} \boldsymbol{\mathcal{A}}_{(k_n)}, \sqrt{\eta^{(t)}} \boldsymbol{U}_{k_n}^{(t)}][\sqrt{\mu^{(t)}} \boldsymbol{\mathcal{B}}_{(k_n)}, \sqrt{\eta^{(t)}} \boldsymbol{I}]^T$: $[\sqrt{\mu^{(t)}} \boldsymbol{\mathcal{A}}_{(k_n)}, \sqrt{\eta^{(t)}} \boldsymbol{U}_{k_n}^{(t)}][\sqrt{\mu^{(t)}} \boldsymbol{\mathcal{B}}_{(k_n)}, \sqrt{\eta^{(t)}} \boldsymbol{I}]^T = \boldsymbol{U}\boldsymbol{\Sigma}\boldsymbol{V}^T$.

**Step 3** Calculate $\boldsymbol{\mathcal{E}}^{(t+1)}$ by

$$\boldsymbol{\mathcal{E}}^{(t+1)} = \arg\min_{\boldsymbol{\mathcal{E}} \in \mathbb{E}} \frac{\mu^{(t)}}{2} \|\Psi_{\mathbb{I}}(\boldsymbol{\mathcal{M}}) - \boldsymbol{\mathcal{X}}^{(t+1)} - \boldsymbol{\mathcal{E}} + \frac{1}{\mu^{(t)}} \boldsymbol{\mathcal{Y}}^{(t)}\|_F^2 + \frac{\eta^{(t)}}{2} \|\boldsymbol{\mathcal{E}} - \boldsymbol{\mathcal{E}}^{(t)}\|_F^2$$

$$= \Psi_{\mathbb{I}^c} \big( \frac{1}{\mu^{(t)} + \eta^{(t)}} (\mu^{(t)} (\Psi_{\mathbb{I}}(\boldsymbol{\mathcal{M}}) - \boldsymbol{\mathcal{X}}^{(t+1)} + \frac{1}{\mu^{(t)}} \boldsymbol{\mathcal{Y}}^{(t)}) + \eta^{(t)} \boldsymbol{\mathcal{E}}^{(t)})), \quad (14)$$

---

**Algorithm 1:** PADMM-based Iterative Solver to (9)

---

**Input:** $\Psi(\mathcal{M})$, $\{\boldsymbol{U}_{k_n}^{(0)}\}_{n=s+1}^{h}$, $\{\hat{\boldsymbol{U}}_{k_n}^{(0)}\}_{n=1}^{s}$, $\mathcal{E}^{(0)}$, $\mathcal{Y}^{(0)}$, $t = 0$, $\rho_\mu, \rho_\eta > 1$, $\bar{\mu}, \bar{\eta}, \mu^{(0)}$, and $\eta^{(0)}$.

**Output:** $\mathcal{Z}^{(t+1)}$, $\{\boldsymbol{U}_{k_n}^{(t+1)}\}_{n=s+1}^{h}$, and

$$\mathcal{X}^{(t+1)} = \mathcal{Z}^{(t+1)} \times_{k_h} \boldsymbol{U}_{k_h}^{(t+1)T} \times_{k_{h-1}} \cdots \times_{k_{s+1}} \boldsymbol{U}_{k_{s+1}}^{(t+1)T}.$$

1. **While not converge do**

2.   Calculate $\mathcal{Z}^{(t+1)}$ by

$$\mathcal{Z}^{(t+1)} = \mathcal{U}^{-1}\Big(\Big(\mathcal{U}\Big(\frac{\mu^{(t)}\hat{\mathcal{P}} \times_{k_{s+1}} \boldsymbol{U}_{k_{s+1}}^{(t)} \cdots \times_{k_h} \boldsymbol{U}_{k_h}^{(t)} + \eta^{(t)}\mathcal{Z}^{(t)}}{\mu^{(t)} + \eta^{(t)}}\Big), \frac{1}{\mu^{(t)} + \eta^{(t)}}\Big)_+\Big);$$

3.   Compute $\boldsymbol{U}_{k_n}^{(t+1)}$ by $\boldsymbol{U}_{k_n}^{(t+1)} = \boldsymbol{U}\boldsymbol{V}^T (n = s+1, \cdots, h)$;

4.   Calculate $\mathcal{X}^{(t+1)}$ by $\mathcal{X}^{(t+1)} = \mathcal{Z}^{(t+1)} \times_{k_h} \boldsymbol{U}_{k_h}^{(t+1)T} \times_{k_{h-1}} \cdots \times_{k_{s+1}} \boldsymbol{U}_{k_{s+1}}^{(t+1)T}$;

5.   Calculate $\mathcal{E}^{(t+1)}$ by

$$\mathcal{E}^{(t+1)} = \Psi_{\mathbb{I}^c}\Big(\frac{1}{\mu^{(t)} + \eta^{(t)}}(\mu^{(t)}(\Psi_{\mathbb{I}}(\mathcal{M}) - \mathcal{X}^{(t+1)} + \frac{1}{\mu^{(t)}}\mathcal{Y}^{(t)}) + \eta^{(t)}\mathcal{E}^{(t)}))\Big);$$

6.   Calculate $\mathcal{Y}^{(t+1)}$ by $\mathcal{Y}^{(t+1)} = \mu^{(t)}(\Psi_{\mathbb{I}}(\mathcal{M}) - \mathcal{X}^{(t+1)} - \mathcal{E}^{(t+1)}) + \mathcal{Y}^{(t)}$;

7.   Compute $\mu^{(t+1)}$ and $\eta^{(t+1)}$ by (16) and (17), respectively;

8.   Check the convergence condition: $\|\mathcal{Z}^{(t+1)} - \mathcal{Z}^{(t)}\|_\infty < \varepsilon$, $\|\mathcal{X}^{(t+1)} - \mathcal{X}^{(t)}\|_\infty < \varepsilon$,
     $\|\boldsymbol{U}_{k_n}^{(t+1)} - \boldsymbol{U}_{k_n}^{(t)}\|_\infty < \varepsilon$ for $n = s+1, s+2, \cdots, h$;

9.   t=t+1.

10. **end while**

---

where $\mathcal{X}^{(t+1)} \times_{k_{s+1}} \boldsymbol{U}_{k_{s+1}}^{(t+1)} \cdots \times_{k_h} \boldsymbol{U}_{k_h}^{(t+1)} = \mathcal{Z}^{(t+1)}$.

**Step 4** Calculate $\mathcal{Y}^{(t+1)}$ by

$$\mathcal{Y}^{(t+1)} = \mu^{(t)}(\Psi_{\mathbb{I}}(\mathcal{M}) - \mathcal{X}^{(t+1)} - \mathcal{E}^{(t+1)}) + \mathcal{Y}^{(t)}. \tag{15}$$

**Step 5** Calculate $\mu^{(t+1)}$ and $\eta^{(t+1)}$ by

$$\mu^{(t+1)} = \min(\bar{\mu}, \rho\mu^{(t)}) \tag{16}$$

and

$$\eta^{(t+1)} = \min(\bar{\eta}, \rho_\eta\eta^{(t)}), \tag{17}$$

where $\rho_\mu, \rho_\eta > 1$, $\bar{\mu}$ and $\bar{\eta}$ are the upper bound of $\mu^{(t+1)}$ and $\eta^{(t+1)}$, respectively.

To solve (9), we repeat Steps 1-5 until convergence is achieved. The entire process is detailed in Algorithm 1.

### 3.2 COMPUTATIONAL COMPLEXITY

Based on the discussions in the previous subsection, the most time-consuming steps in the algorithm 1 are the computations of $\mathcal{Z}$, and $\boldsymbol{U}_{k_n}$. Since the computational complexity of n-mode product of $\hat{\mathcal{P}} \in \mathbb{R}^{I_1 \times I_2 \times \cdots \times I_h}$ and $\boldsymbol{U}_n \in \mathbb{R}^{I_n \times I_n}$ is $\mathcal{O}(I_n I_1 I_2 \cdots I_h)$, the computational complexity of the computation of $\mathcal{Z}$ is $\mathcal{O}(h I_{(1)} I_1 I_2 \cdots I_h)$, where $\max_k(I_k) = I_{(1)}$. Besides, the computational complexity of the computation of $\boldsymbol{U}_{k_n}$ is $\mathcal{O}((h-1)I_{(1)}I_1 I_2 \cdots I_h + I_k^2 I_1 I_2 \cdots I_h)$, therefore the overall computational complexity of each iteration of Algorithm 1 is $\mathcal{O}\Big((h-s)(h + I_{(1)} - 1)I_{(1)}I_1 I_2 \cdots I_h + h I_{(1)} I_1 I_2 \cdots I_h\Big)$.

### 3.3 CONVERGENCE ANALYSIS

Although the optimization problem (9) is non-convex because of the constraints $\boldsymbol{U}_{k_n}^T \boldsymbol{U}_{k_n} = \boldsymbol{I}(n = s+1, s+2, \cdots, h)$ and the global optimality for (9) is hardly guaranteed, we can still prove some excellent convergence properties of the Algorithm 1, as stated in the following theorem.

**Theorem 1.** *For the sequence* $\{[\mathcal{Z}^{(t)}, \{\boldsymbol{U}_{k_n}^{(t)}\}_{n=s+1}^{h}, \mathcal{E}^{(t)}, \mathcal{Y}^{(t)}, \mu^{(t)}]\}$ *generated by the proposed algorithm 1, we have the following properties if* $\{\mathcal{Y}^{(t)}\}$ *is bounded,* $\sum_{t=1}^{\infty}(\mu^{(t)})^{-2}\mu^{(t+1)} < +\infty$ *and* $\lim_{n \to \infty} \mu^{(n)} \sum_{t=n}^{\infty}(\eta^{(t)})^{-1/2} = 0$.

*(i)* $\lim\limits_{t \longrightarrow \infty} \Psi_{\mathbb{I}}(\mathcal{M}) - \mathcal{X}^{(t)} - \mathcal{E}^{(t)} = \mathbf{0}$, *where* $\mathcal{X}^{(t)} = \mathcal{Z}^{(t)} \times_{k_h} (\boldsymbol{U}_{k_h}^{(t)})^T \cdots \times_{k_{s+1}} (\boldsymbol{U}_{k_{s+1}}^{(t)})^T$.

*(ii)* $\{[\mathcal{Z}^{(t)}, \{\boldsymbol{U}_{k_n}^{(t)}\}_{n=s+1}^h], \mathcal{X}^{(t)}, \mathcal{E}^{(t)}]\}$ *is bounded.*

*(iii)* $\sum_{t=1}^{\infty} \eta^{(t)} \|[\mathcal{Z}^{(t)}, \{\boldsymbol{U}_{k_n}^{(t)}\}_{n=s+1}^h], \mathcal{E}^{(t)}] \quad - \quad [\mathcal{Z}^{(t+1)}, \{\boldsymbol{U}_{k_n}^{(t+1)}\}_{n=s+1}^h], \mathcal{E}^{(t+1)}]\|_F^2$ *is convergent. Thus, we have* $\|[\mathcal{Z}^{(t)}, \{\boldsymbol{U}_{k_n}^{(t)}\}_{n=s+1}^h], \mathcal{E}^{(t)}] \quad - \quad [\mathcal{Z}^{(t+1)}, \{\boldsymbol{U}_{k_n}^{(t+1)}\}_{n=s+1}^h], \mathcal{E}^{(t+1)}]\|_F^2 \leq \mathcal{O}(\frac{1}{\eta^{(t)}})$.

*(iv)* $\lim\limits_{t \longrightarrow \infty} \|\mathcal{Y}^{(t+1)} - \mathcal{Y}^{(t)}\|_F = 0$.

*(v)* *Let* $[\mathcal{Z}^*, \{\boldsymbol{U}_{k_n}^*\}_{n=s+1}^h, \mathcal{E}^*, \mathcal{Y}^*]$ *be any limit point of* $\{[\mathcal{Z}^{(t)}, \{\boldsymbol{U}_{k_n}^{(t)}\}_{n=s+1}^h, \mathcal{E}^{(t)}, \mathcal{Y}^{(t)}]\}$. *Then,* $[\mathcal{Z}^*, \{\boldsymbol{U}_{k_n}^*\}_{n=s+1}^h, \mathcal{E}^*, \mathcal{Y}^*]$ *is a KKT point to* (9).

Please refer to the supplementary material of this paper for the proof of the Theorem 1. Theorem 1 shows that, if $\{\mathcal{Y}^{(t)}\}$ is bounded, the sequence $[\mathcal{Z}^{(t)}, \{\boldsymbol{U}_{k_n}^{(t)}\}_{n=s+1}^h, \mathcal{E}^{(t)}]$ generated by the proposed algorithm 1 is Cauchy convergent, with a convergence rate of at least $\mathcal{O}(\frac{1}{\eta^{(t)}})$. Moreover, any accumulation point of the sequence converges to the KKT point of (9).

## 4 STABLE TC-U1

**Considering the case that the tensor data may contaminated by noise, we utilize the proposed TDST to derive the following model (STC-U1) based on the given $\{\hat{\boldsymbol{U}}_{k_n}\}_{n=3}^s$:**

$$\min_{\mathcal{X}, \boldsymbol{U}_{k_n}^T \boldsymbol{U}_{k_n} = \boldsymbol{I}(n=s+1,\cdots,h)} \|\mathcal{X} \times_{k_{s+1}} \boldsymbol{U}_{k_{s+1}} \cdots \times_{k_h} \boldsymbol{U}_{k_h}\|_{\mathcal{U},1}$$
$$s.t. \ \|\Psi_{\mathbb{I}}(\mathcal{M}) - \Psi_{\mathbb{I}}(\mathcal{X})\|_F \leq \delta, \tag{18}$$

**where $\delta \geq 0$ represents the magnitude of the noise.**

**Lemma 1.** *For* $\mathcal{A} \in \mathbb{R}^{I_1 \times I_2 \times \cdots \times I_h}$, *the subgradient of* $\|\mathcal{A}\|_{1,\mathcal{U}}$ *is given as* $\partial \|\mathcal{A}\|_{1,\mathcal{U}} = \{\mathcal{U}^{-1}(\text{sgn}(\mathcal{U}(\mathcal{A})) + \mathcal{F}|\Psi_{\hat{\mathbb{H}}}(\mathcal{U}(\mathcal{F})) = \mathbf{0}, \|\mathcal{F}\|_{\mathcal{U},\infty} \leq 1\}$, *where* $\hat{\mathbb{H}}$ *denotes the support of* $\mathcal{U}(\mathcal{A})$.

**Let $(\hat{\mathcal{X}}, \{\hat{U}_k\}_{k=1}^h)$ be the result by (18) and $\mathbb{H}$ be the support of $\hat{\mathcal{U}}(\mathcal{M})$. Defining $P_{\mathbb{S}}(\mathcal{A}) = \Psi_{\mathbb{H}}(\hat{\mathcal{U}}(\mathcal{A})) = \mathcal{A} - P_{\mathbb{S}^\perp}(\mathcal{A})$ for any $\mathcal{A} \in \mathbb{R}^{I_1 \times I_2 \times \cdots \times I_h}$, we have the following result.**

**Theorem 2.** *If the dual certificate* $\mathcal{G} = \Psi_{\mathbb{I}} P_{\mathbb{S}} (P_{\mathbb{S}} \Psi_{\mathbb{I}} P_{\mathbb{S}})^{-1} (\hat{\mathcal{U}}^{-1}(\text{sgn}(\hat{\mathcal{U}}(\mathcal{M}))))$ *satisfies* $\|P_{\mathbb{S}^\perp}(\mathcal{G})\|_{\hat{\mathcal{U}},\infty} \leq C_1 < 1$ *and* $P_{\mathbb{S}} \Psi_{\mathbb{I}} P_{\mathbb{S}} \succeq C_2 p \mathcal{I}$, *then we can obtain the following inequality:*

$$\|\mathcal{M} - \hat{\mathcal{X}}\|_F \leq \frac{1}{1 - C_1} \sqrt{\frac{1/C_2 + p}{p} I_1 I_2} \delta + \delta, \tag{19}$$

*where $\hat{\mathcal{X}}$ is obtained by* (18) *and $p$ denotes the sampling rate.*

**From the theorem 2, we obtain a theoretical guarantee for the exact recovery of TC-U1 when $\delta = 0$. Similar results for TC-SL can be found in the supplementary material.**

## 5 EXPERIMENTAL RESULTS

In this section, we conducted two sets of experiments to illustrate the effectiveness of our proposed method in high-order tensor completion: in the first set, we ran a series of numerical experiments on tensors with $(1 - p)I_1 \cdots I_h$ missing entries **to investigate the exact recovery capability of TC-U1 in a more intuitive manner**; in the second set, we compared the TC-SL and TC-U1 [2] with

---

[2]**For real-world applications, the parameters in TC-SL and TC-U1 were configured as $\{s = 2, (k_3, k_4) = (1, 4)\}$ and $\{s = 2, (\hat{\boldsymbol{U}}_{k_1}, \hat{\boldsymbol{U}}_{k_2}) = (\boldsymbol{F}_{I_1}, \boldsymbol{F}_{I_2})\}$, respectively. The configurations are based on an assumption that the first two dimensions of images exhibit smoothness characteristics and similarity of different channel of images.**

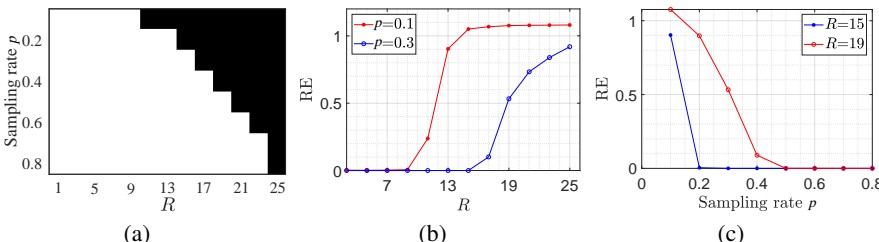

Figure 1: Recovery performance of TC-U1 on the synthetic data with varying $(R, p)$ and size of $30 \times 30 \times 30 \times 30$.

several state-of-the-art methods, including TNN-DCT Lu et al. (2019b), TNN-DFT Lu et al. (2018), SNN Liu et al. (2013), KBR Xie et al. (2017), WSTNN Zheng et al. (2020), and HTNN-DCT Qin et al. (2022), in the context of image sequences inpainting and color video inpainting. To ensure the reliability of our experiments, we conducted each experiment five times and reported the average results as the final outcomes. All experiments were implemented using Matlab R2022b.[3]

## 5.1 SYNTHETIC EXPERIMENTS

For given **rank parameter** $R$, we generated a $h$-order tensor data with size of $(I_1, I_2, \cdots, I_h)$ as $\boldsymbol{\mathcal{M}} = \boldsymbol{\mathcal{G}}_0 \times_h \boldsymbol{U}_h \times_{h-1} \cdots \times_1 \boldsymbol{U}_1$. Here, $\boldsymbol{\mathcal{G}}_0 \in \mathbb{R}^{R \times R \times \cdots \times R}$ is i.i.d. sampled from a normal distribution $\mathcal{N}(0, 1)$ and $\boldsymbol{U}_k \in \mathbb{R}^{I_k \times R}$ is matrix obtained by randomly selecting $R$ columns from Discrete Cosine Matrix (DCM). We use the relative error (RE) defined as $||\boldsymbol{\mathcal{M}} - \hat{\boldsymbol{\mathcal{M}}}||_F / ||\boldsymbol{\mathcal{M}}||_F$ to measure estimation errors, where $\hat{\boldsymbol{\mathcal{M}}}$ is the recovered tensor by our method. All results are presented in Fig. 1. In the Fig. 1 (a), a successful trial, *i.e.*, $\text{RE} \leq 10^{-2}$, is presented in white region. From the figure, it is evident that TC-U1 can accurately recover the true tensor when the values of $p$ and $R$ satisfy a fairly broad condition. Furthermore, in Fig. 1(b), we observe that as the value of $R$ decreases for a fixed $p$, the RE tends to approach zero rapidly. This indicates that TC-U1 is capable of achieving high accuracy in tensor recovery for the case of small $R$. Besides, in Figure 1(c), the RE remains stable for a fixed value of $R$ as the sampling rate varies. This stability further highlights the effectiveness of TC-U1 in practical scenarios.

## 5.2 REAL-WORLD APPLICATIONS

In this subsection, we evaluated different tensor completion methods using three image sequence databases: *Berkeley Segmentation Dataset (BSD)* Martin et al. (2001), *Unfiltered Faces for Gender and Age Classification (UFGAC)*, *Labeled Faces in the Wild (LFW)*, and *Georgia Tech Face database (GTF)*[4], as well as the color video database *HMDB51*[5] (the comparison on *HMDB51* is given in the supplementary material of this paper). The observation tensors for the three-order tensor methods (TNN-DCT and TNN-DFT) were obtained by the mode-12 unfolding tensor, *i.e.*, $[\Psi_{\mathbb{I}}(\boldsymbol{\mathcal{M}})]_{(12)} \in \mathbb{R}^{I_1 \times I_2 \times I_3 I_4}$. Here, $I_1 \times I_2$ represents the size of each image (frame), $I_3 = 3$ is RGB channel number of each image, and $I_4$ is the number of images (frames). Random shuffling was applied to the image sequences to ensure unbiased evaluation before testing. We used the Peak Signal-To-Noise Ratio (PSNR) to evaluate the performance of different methods in tensor completion. The best results for each case are shown in bold.

Table 3 presents the PSNR values achieved by each method on the image sequence databases with different scenes. From the table, most of the higher tensor methods (KBR, WSTNN, HTNN-

---

[3]For a fair comparison, we implemented the compared methods using the code provided by the respective authors in our experimental environment.The code for our proposed method is available at the following link: `https://github.com/nobody111222333/MatlabCode.git`.

[4]Specifically, for the BSD database (`https://www2.eecs.berkeley.edu/Research/Projects/CS/vision/bsds/`), we randomly selected 50 color images. For UFGAC (`https://talhassner.github.io/home/projects/Adience/Adience-data.html`), we chose all color images from the first class. For LFW (`http://vis-www.cs.umass.edu/lfw/`), we selected the first 50 classes. For GTF (`http://www.anefian.com/research/face_reco.htm`), we selected the first five classes.

[5]https://serre-lab.clps.brown.edu/resource/hmdb-a-large-human-motion-database/

Table 3: Comparing various methods on *BSD* at different sampling rates.

| Sampling Rate $p$ | TNN-DCT | TNN-DFT | SNN | KBR | WSTNN | HTNN-DCT | TC-SL | TC-U1 |
|---|---|---|---|---|---|---|---|---|
| 0.3 | 23.25 | 23.21 | 21.86 | 25.45 | 25.75 | 25.21 | 26.32 | **26.95** |
| 0.5 | 27.25 | 27.20 | 25.50 | 31.57 | 31.07 | 30.72 | 31.55 | **34.53** |
| 0.7 | 32.04 | 31.95 | 29.84 | 38.81 | 37.11 | 38.22 | 38.33 | **41.76** |

Table 4: Comparing various methods on the three datasets at sampling rates $p = 0.3$.

| Data | TNN-DCT | TNN-DFT | SNN | KBR | WSTNN | HTNN-DCT | TC-SL | TC-U1 |
|---|---|---|---|---|---|---|---|---|
| UFGAC | 36.42 | 36.36 | 36.94 | 41.58 | 34.13 | 30.84 | 41.27 | **44.42** |
| GTF | 25.46 | 25.41 | 23.54 | 25.83 | 26.66 | 22.11 | 32.10 | **32.85** |
| LFW | 28.14 | 28.10 | 22.75 | 33.86 | 35.10 | 31.28 | 32.06 | **36.91** |
| Average | 30.01 | 29.96 | 27.74 | 33.76 | 31.96 | 28.08 | 35.14 | **38.06** |

DCT, TC-SL, and TC-U1) achieved better performance for all cases compared with the three-order tensor methods (TNN-DCT and TNN-DFT). This indicates that high-order tensor methods are more effective in accurately capturing the low-rank structure in four-order tensor data. **Notably, TC-U1 outperforms the other methods by about 3 dB for $p \in \{0.5, 0.7\}$ and demonstrates particularly impressive improvements compared to the other higher tensor methods.** Furthermore, in Table 4, we compared the tensor recovery performance of all methods on the cases when the tensor data suffer the random slices permutations. Notably, both TC-SL and TC-U1 achieve the best performance for all cases. **In particular, the average PSNR results obtained by TC-SL and TC-U1 outperform the other methods by more than 1 dB and 4 dB, respectively.** In the case of *GTF*, TC-U1 even outperforms WSTNN by more than 6 dB and surpasses the other methods by more than 7 dB in terms of PSNR!

These results suggest that the proposed TC-U1 is able to exploit the correlations of tensor data along different dimensions effectively and can more accurately exploit the low-rank structure of four-order tensor data (the one with non-continue change) than other methods. And the superiority of TC-SL and TC-U1 over other methods can be attributed to the introduction of both the given transforms $\{\hat{\boldsymbol{U}}_{k_n}\}_{n=1}^{s}$ from smooth priors and learnable unitary matrices $\{\boldsymbol{U}_{k_n}\}_{n=s+1}^{h}$, which enable it to better handle the non-smooth in tensor data caused by random shuffling image sequences (such as *UFGAC*, *LFW*, and *GTF*) or the concatenation of different scene images (such as *BSD*) and capture the underlying low-rank structures in the tensor data more effectively.

## 6 CONCLUSIONS

This paper studied a high-order tensor completion problem that aims to more accurately recover the true low-rank tensor from the observed tensor with missing elements and non-smooth change. To achieve this goal, we introduced a new tensor decomposition named TDSL and a new norm named tensor $U_1$ norm successively. The PADMM is proposed for solving the proposed two tensor completion models (including TC-SL and TC-U1). The both theoretical analysis and synthetic experiments provided strong evidence for the exact recovery capabilities of the proposed tensor completion methods under a fairly broad condition, supporting the applicability of the proposed method to practical problems involving tensor recovery. The experimental results in real-world applications showcased the superiority of our methods over previous methods in high-order tensor completion, especially for the tensor data with non-smooth changes. **Additionally, thanks to the proposed methods, we no longer need to introduce $\binom{h}{2}$ variables and tune weighted parameters to study the correlation information of the tensor across different dimensions. It is worth noting that the tensor decomposition methods and norms we propose can extend beyond tensor completion alone and are applicable to diverse tasks, such as tensor robust PCA, stable tensor completion, and tensor low-rank representation.**

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
