# Supplementary Material of High-Order Tensor Recovery with A Tensor $U_1$ Norm

## 1 Related Works

Unlike the well-defined matrix rank, there is currently no universally accepted definition for tensor rank. In this part, we will briefly introduce three common definitions of tensor rank based on different decomposition techniques: Canonical Polyadic (CP) Decomposition Hitchcock (1927; 1928), Tucker Decomposition Tucker (1963), and methods based on t-SVD Lu et al. (2019b; 2018); Qin et al. (2022); Wang et al. (2023); Zhang et al. (2022); Zhou et al. (2017).

It is worth noting that from an equivalent definition of matrix rank, a rank $r$ matrix can be written as the sum of $r$ rank-one matrix. Inspired by that, Kolda and Bader Kolda & Bader (2009b) have proposed CP rank, *i.e.*, $\mathrm{rank}_{\mathrm{cp}}(\cdot)$, defined on tensor rank-one decomposition (CP Decomposition):

$$\mathrm{rank}_{\mathrm{cp}}(\boldsymbol{\mathcal{X}}) = \min\{R | \boldsymbol{\mathcal{X}} = \sum_{r=1}^{R} g_{r,r,\cdots,r} \boldsymbol{u}_r^{(1)} \circ \boldsymbol{u}_r^{(2)} \circ \cdots \circ \boldsymbol{u}_r^{(h)}, \boldsymbol{u}_r^{(j)} \in \mathbb{R}^{I_j} \text{ for } j = 1, 2, \cdots, h\} \quad (1)$$

for tensor $\boldsymbol{\mathcal{X}} \in \mathbb{R}^{I_1 \times I_2 \times \cdots \times I_h}$. We can see from (1) that the definition of matrix rank is a special case of CP rank. But solving (1) is time-consuming even for small tensor when $h \geq 3$.

As the computation of CP rank is NP-hard and greatly restricts its application in tensor recovery, various Tucker Decomposition-based methods for defining tensor rank have been proposed and extensively studied than CP rank Kolda & Bader (2009b). Given $\boldsymbol{\mathcal{A}} \in \mathbb{R}^{I_1 \times I_2 \times \cdots \times I_h}$, the Tucker Decomposition of $\boldsymbol{\mathcal{A}}$ is written as $\boldsymbol{\mathcal{A}} = \boldsymbol{\mathcal{G}} \times_1 \boldsymbol{U}_1 \times_2 \boldsymbol{U}_2 \times_3 \cdots \times_h \boldsymbol{U}_h$, where $\boldsymbol{\mathcal{G}} \in \mathbb{R}^{R_1 \times R_2 \times \cdots R_h}$, and $\boldsymbol{U}_k \in \mathbb{R}^{I_k \times R_k}$ for $k = 1, 2, \cdots, h$. Given $\mathrm{rank}(\boldsymbol{A}_{(k)})$ for all $k$, we can obtain the decomposition by the higher-order singular value decomposition (HOSVD) De Lathauwer et al. (2000), where $\mathrm{rank}(\boldsymbol{A}_{(k)}) = R_k$ for $k = 1, 2, \cdots, h$. Therefore, the Tucker rank of tensor $\boldsymbol{\mathcal{A}}$ is defined as

$$\mathrm{rank}_{\mathrm{tc}}(\boldsymbol{\mathcal{A}}) = (\mathrm{rank}(\boldsymbol{A}_{(1)}), \mathrm{rank}(\boldsymbol{A}_{(2)}), \cdots, \mathrm{rank}(\boldsymbol{A}_{(h)}),$$

which is also known as multilinear rank and n-rank. Based on the Tucker rank, Gandy *et al.* given a new rank of the tensor data defined as $\sum_{n=1}^{h} \mathrm{rank}(\boldsymbol{A}_{(n)})$ Gandy et al. (2011). Furthermore, considering the difference of the low rankness along different dimensions of tensor data, Liu et al. (2013) give a weighted sum of the ranks of the unfolding matrices $\sum_{n=1}^{h} \alpha_n \mathrm{rank}(\boldsymbol{A}_{(n)})$, where the weights $\alpha_n(n = 1, 2, \cdots, h)$ satisfy $\sum_{n=1}^{h} \alpha_n = 1$ and play an important role in the newly defined rank. However, the best choice for the weights is hard to know if without any prior. Thus, a new tensor rank based on the maximum rank of a set of unfolding matrics is proposed to promote the low-rankness of unfolding matrics of the recovered tensor Zhang et al. (2014a).

Recently, there has been a growing interest in tensor rank by using t-SVD Zhang et al. (2014b); Lu et al. (2019a;b). This approach defines rank based on the Singular Value Decomposition (SVD) of frontal slices of the tensor resulting from invertible transforms applied along a specific dimension (known as t-SVD). This approach is widely employed in tensor recovery, as they can better utilize the smoothness priors in tensor data due to the use of transforms such as DFT. For example, Zhang et al. (2014b) introduced a tensor tubal rank based on the Discrete Fourier Transform (DFT) for three-order tensors. It counts the number of non-zero tensor tubes in the singular value tensor obtained by performing frontal-slices-wise SVD of the transformed tensor. Similarly, Lu et al. (2019a) defined tensor average rank for three-order tensors based on DFT, which averages the ranks of frontal slices of the transformed tensor and provided theoretical guarantees for exact recovery using the convex hull of tensor average rank. As noted in Lu et al. (2019a), the low tensor average rank assumption for tensor data can be more easily satisfied in the real world than the low-rank assumption employed in the

**Algorithm 2:** Tensor Decomposition Based on Slices-Wise Low-Rank Prior (TDSL)

---

**Input:** $\mathcal{A} \in \mathbb{R}^{I_{k_1} \times I_{k_2} \times \cdots \times I_{k_h}}$, $\{\hat{U}_{k_n}\}_{n=3}^{s}$, and $r$, where $1 \le k_i \ne k_j (\text{if } i \ne j) \le h$

**Output:** $\mathcal{Z}_1, \{U_{k_n}\}_{n=s+1}^{h}$.

1. $\bar{\mathcal{A}} = \mathcal{A} \times_{k_3} \hat{U}_{k_3} \cdots \times_{k_s} \hat{U}_{k_s}$

**while** not converged **do**

2. Calculate the slices-wise SVD for $\bar{\mathcal{A}} \times_{k_{s+1}} U_{k_{s+1}}^{(t)} \cdots \times_{k_h} U_{k_h}^{(t)}$ by computing SVD of its all
   slices along the $(k_1, k_2)$-th mode: for all $1 \le i_{k_3} \le I_{k_3}, ..., 1 \le i_{k_h} \le I_{k_h}$, we have
   $[\bar{\mathcal{A}} \times_{k_{s+1}} U_{k_{s+1}}^{(t)} \cdots \times_{k_h} U_{k_h}^{(t)}]_{:,:,i_{k_3},\cdots,i_{k_h}} = [\bar{\mathcal{U}}]_{:,:,i_{k_3},\cdots,i_{k_h}} [\bar{\mathcal{S}}]_{:,:,i_{k_3},\cdots,i_{k_h}} [\bar{\mathcal{V}}]_{:,:,i_{k_3},\cdots,i_{k_h}}^T.$

3. Calculate $\mathcal{Z}_1^{(t+1)}$ by $[\mathcal{Z}_1]_{:,:,i_{k_3},\cdots,i_{k_h}}^{(t+1)} = [\bar{\mathcal{U}}]_{:,1:r,i_{k_3},\cdots,i_{k_h}} [\bar{\mathcal{S}}]_{1:r,1:r,i_{k_3},\cdots,i_{k_h}} [\bar{\mathcal{V}}]_{:,1:r,i_{k_3},\cdots,i_{k_h}}^T$

4. Compute $U_{k_n}^{(t+1)}$ for all $s + 1 \le n \le h$ by $U_{k_n}^{(t+1)} = UV^T$, where $U$ and $V$ are obtained by
   SVD for $[\mathcal{Z}_1]_{(k_n)} \mathcal{Y}_{(k_n)}^T$, $i.e.$, $[\mathcal{Z}_1]_{(k_n)} \mathcal{Y}_{(k_n)}^T = USV^T$, and
   $\mathcal{Y} = \bar{\mathcal{A}} \times_{k_{s+1}} U_{k_{s+1}}^{(t+1)} \cdots \times_{k_{n-1}} U_{k_{n-1}}^{(t+1)} \times_{k_{n+1}} U_{k_{n+1}}^{(t)} \cdots \times_{k_h} U_{k_h}^{(t)}.$

3. Check the convergence conditions: $\|\mathcal{Z}_1^{(t+1)} - \mathcal{Z}_1^{(t)}\|_\infty < \varepsilon$, $\|U_{k_n}^{(t+1)} - U_{k_n}^{(t)}\|_\infty < \varepsilon$ for all
   $s + 1 < n \le h$;

4. $t = t + 1$.

**end while**

---

tensor tubal rank, CP rank, and tucker rank. Specifically, the tensor average rank of any three-order tensor $\mathcal{A}$ satisfied the following inequation

$$\text{rank}_a(\mathcal{A}) \le \max \text{rank}_{tc}(\mathcal{A}) \le \text{rank}_{cp}(\mathcal{A}), \tag{2}$$

where $\text{rank}_{tc}(\mathcal{A})$ and $\text{rank}_{cp}(\mathcal{A})$ are the Tucker rank Kolda & Bader (2009a) and CP rank Kolda & Bader (2009b) of $\mathcal{A}$, respectively. Employing a similar idea to tensor average rank, a new rank based on real invertible transforms has been given in Lu et al. (2019b), and defined as $\text{rank}_L(\mathcal{A}) = \frac{1}{I_3} \sum_{i_3=1}^{I_3} \text{rank}([\mathcal{A} \times_3 L]_{:,:,i_3})$, where $L$ is a fixed real invertible transform, such as Discrete Cosine Matrix (DCM) and Random Orthogonal Matrix (ROM), that satisfies $L^T L = LL^T = \ell_L$, and $\ell_L$ is a constant. To handle the higher order tensor case, in Qin et al. (2022), the slice-wise low rankness of $\mathcal{L}(\mathcal{A})$ is considered, where $\mathcal{L}(\mathcal{A}) = \mathcal{X} \times_3 L_3 \times_4 \cdots \times_h L_h$, $\mathcal{L}^T(\mathcal{A}) = \mathcal{X} \times_h L_h^T \times_{h-1} \cdots \times_3 L_3^T$, and $\mathcal{L}^T(\mathcal{L}(\mathcal{I})) = \mathcal{L}(\mathcal{L}^T(\mathcal{I})) = \ell_{\mathcal{L}} \mathcal{I}$ for given invertible transforms $\{L_k\}_{k=3}^{h}$. Considering the difference of tensor low-rankness across different dimensions of the tensor, Zheng et al. (2020) give WSTNN, which is defined as the weighted sum of the tensor average rank of all $\binom{h}{2}$ mode-$k_1 k_2$ unfolding tensor. However, it will become impractical as the tensor order $h$ increases. Besides, the weight parameter tuning can also be a challenge. These t-SVD-based methods utilize the smoothness priors in tensor data better than the other methods due to the use of transforms such as DFT, but it is also exactly why they are sensitive to non-smooth changes and slice permutations of tensor data. Zheng et al. (2022) proposed a solution to address the slice permutation issue in DFT-based methods by minimizing a Hamiltonian circle, though it is limited to DFT. Moreover, the methods based on t-SVD introduce more variables and weight parameters compared to CP and Tucker rank methods.

## 2    TDSL (ALGORITHM 2)

## 3    THE PROOF OF PROPERTY 2

*Proof.* (i) We can conclude that both the tensor $U_1$ norm and tensor $U_\infty$ norm are convex due to the convexity properties of the $l_1$-norm and $\infty$-norm, respectively.

(ii)

$$\sup_{\|\mathcal{B}\|_{\mathcal{U},\infty} \le 1} \langle \mathcal{A}, \mathcal{B} \rangle = \sup_{\|\mathcal{B} \times_1 \hat{U}_1 \times_2 \cdots \times_h \hat{U}_h\|_\infty \le 1} \langle \mathcal{A}, \mathcal{B} \rangle$$

$$= \sup_{\|\mathcal{B} \times_1 \hat{U}_1 \times_2 \cdots \times_h \hat{U}_h\|_\infty \le 1} \left\langle \mathcal{A} \times_1 \hat{U}_1 \times_2 \cdots \times_h \hat{U}_h, \mathcal{B} \times_1 \hat{U}_1 \times_2 \cdots \times_h \hat{U}_h \right\rangle.$$

Let $\hat{\mathcal{B}} = \mathcal{B} \times_1 \hat{U}_1 \times_2 \cdots \times_h \hat{U}_h$ be any tensor. Then we have

$$\sup_{\|\hat{\mathcal{B}}\|_\infty \leq 1} \left\langle \mathcal{A} \times_1 \hat{U}_1 \times_2 \cdots \times_h \hat{U}_h, \hat{\mathcal{B}} \right\rangle = \|\mathcal{A} \times_1 \hat{U}_1 \times_2 \cdots \times_h \hat{U}_h\|_1 = \|\mathcal{A}\|_{\mathcal{U},1}. \qquad (3)$$

(iii) The proof is completed in the following two steps, utilizing the properties of conjugate functions presented in Fazel Sarjoui (2002); Hiriarturruty & Lemaréchal (1993), *i.e.*, the conjugate of the conjugate, $\phi_0^{**}$, is the convex envelope of a given function $\phi_0 : \mathbb{C} \to \mathbb{R}$. For given function $\phi_0$, the conjugate $\phi_0^*$ of the function $\phi_0$ is defined as $\phi_0^*(y) = \sup\{\langle y, x \rangle - \phi_0(x) | x \in \mathbb{C}\}$.

STEP1. COMPUTING THE CONJUGATE OF SPARSITY-BASED TENSOR $U_0$, $\phi^*$.

$$\phi^*(\mathcal{B}) = \sup_{\mathcal{A} \in \mathbb{S}} \langle \mathcal{B}, \mathcal{A} \rangle - \|\mathcal{A}\|_{\mathcal{U},0} = \sup_{\|\mathcal{A}\|_{\mathcal{U},\infty} \leq 1} \langle \mathcal{B}, \mathcal{A} \rangle - \|\mathcal{A} \times_1 \hat{U}_1 \times_2 \cdots \times_h \hat{U}_h\|_0$$

$$(\text{Let } \hat{\mathcal{A}} = \mathcal{A} \times_1 \hat{U}_1 \times_2 \cdots \times_h \hat{U}_h \text{ be any tensor.})$$

$$= \sup_{\|\hat{\mathcal{A}}\|_\infty \leq 1} \left\langle \mathcal{B} \times_1 \hat{U}_1 \times_2 \cdots \times_h \hat{U}_h, \hat{\mathcal{A}} \right\rangle - \|\hat{\mathcal{A}}\|_0$$

$$= \begin{cases} 0, \|\mathcal{B}\|_{\mathcal{U},\infty} \leq 1; \\ \|(\mathcal{B} \times_1 \hat{U}_1 \times_2 \cdots \times_h \hat{U}_h, 1)_+\|_1, \text{ otherwise.} \end{cases}$$

STEP2. COMPUTING THE CONJUGATE OF $\phi^*$, $\phi^{**}$.   Defining

$$f(\mathcal{A}_0) = \begin{cases} 0, \|\mathcal{A}_0\|_\infty \leq 1; \\ \|(\mathcal{A}_0, 1)_+\|_1, \text{ otherwise,} \end{cases}$$

we have

$$\phi^{**}(\mathcal{C}) = \sup_{\mathcal{B}} \langle \mathcal{C}, \mathcal{B} \rangle - \phi^*(\mathcal{B}) = \sup_{\mathcal{B}} \left\langle \mathcal{C} \times_1 \hat{U}_1 \times_2 \cdots \times_h \hat{U}_h, \mathcal{B} \times_1 \hat{U}_1 \times_2 \cdots \times_h \hat{U}_h \right\rangle - \phi^*(\mathcal{B})$$

$$(\text{Let } \hat{\mathcal{B}} = \mathcal{B} \times_1 \hat{U}_1 \times_2 \cdots \times_h \hat{U}_h \text{ be any tensor.})$$

$$= \sup_{\hat{\mathcal{B}}} \left\langle \mathcal{C} \times_1 \hat{U}_1 \times_2 \cdots \times_h \hat{U}_h, \hat{\mathcal{B}} \right\rangle - f(\hat{\mathcal{B}}) = \|\mathcal{C}\|_{\mathcal{U},1}$$

over the set $\mathbb{S}$.

$\square$

## 4   THE PROOF OF THEOREM 1

Without loss of generality, let us consider

$$\min_{\mathcal{Z}, U_k^T U_k = I(k=s+1,\cdots,h)} \|\mathcal{Z}\|_{\mathcal{U},1} \qquad s.t. \ \Psi_\mathbb{I}(\mathcal{M}) = \mathcal{Z} \times_{s+1} U_{s+1}^T \cdots \times_h U_h^T + \mathcal{E}, \qquad (4)$$

where $\mathcal{U}(\mathcal{Z}) = \mathcal{Z} \times_1 U_1 \cdots \times_s U_s$.

$$\mathcal{L}_a(\mathcal{Z}, \{U_k\}_{k=s+1}^h, \mathcal{E}, \mathcal{Y}, \{Y_k\}_{k=s+1}^h, \mathcal{W})$$

$$= \|\mathcal{Z}\|_{\mathcal{U},1} + \left\langle \Psi_\mathbb{I}(\mathcal{M}) - \mathcal{Z} \times_h U_h^T \times_2 \cdots \times_{s+1} U_{s+1}^T - \mathcal{E}, \mathcal{Y} \right\rangle + \sum_{k=s+1}^h \left\langle U_k^T U_k - I, Y_k \right\rangle + \langle \Psi_\mathbb{I}(\mathcal{E}), \mathcal{W} \rangle$$

$$(5)$$

From (5), *i.e.*, the Lagrangian function of (4), we can get the following KKT conditions by the first order optimality conditions for (4):

$$\begin{cases} \Psi_\mathbb{I}(\mathcal{M}) - \mathcal{X} - \mathcal{E} = 0; \\ \mathcal{Y} \times_{s+1} U_{s+1} \times_{s+2} \cdots \times_h U_h \in \partial \|\mathcal{Z}\|_{\mathcal{U},1}; \\ U_k^T U_k = I \text{ for } k = s+1, s+2, \cdots h \\ -\mathcal{F}_{(k)}(\mathcal{C}_{(k)})^T + U_k(Y_k + Y_k^T) = 0; \\ \Psi_\mathbb{I}(\mathcal{E}) = 0; \\ -\Psi_{\mathbb{I}^c}(\mathcal{Y}) = 0; \\ -\Psi_\mathbb{I}(\mathcal{Y}) + \mathcal{W} = 0, \end{cases} \qquad (6)$$

where $\mathcal{C} = \mathcal{Y} \times_{s+1} U_{s+1} \cdots \times_{k-1} U_{k-1}$ and $\mathcal{F} = \mathcal{Z} \times_h (U_h)^T \cdots \times_{k+1} (U_{k+1})^T$.

*Proof.* (i) By $\Psi_{\mathbb{I}}(\mathcal{M}) - \mathcal{X}^{(t+1)} - \mathcal{E}^{(t+1)} = (\mu^{(t)})^{(-1)}(\mathcal{Y}^{(t+1)} - \mathcal{Y}^{(t)})$ and the boundedness of $\mathcal{Y}^{(t)}$, we have $\lim_{t \longrightarrow \infty} \Psi_{\mathbb{I}}(\mathcal{M}) - \mathcal{X}^{(t+1)} - \mathcal{E}^{(t+1)} = \mathbf{0}$.

(ii) From the the optimality of $\mathcal{Z}^{(t+1)}$, $\{U_k^{(t+1)}\}_{k=s+1}^h$, and $\mathcal{E}^{(t+1)}$, we have

$$
\begin{aligned}
&\mathcal{L}(\mathcal{Z}^{(t+1)}, \{U_k^{(t+1)}\}_{k=s+1}^h, \mathcal{E}^{(t+1)}, \mathcal{Y}^{(t)}, \mu^{(t)}) \\
\leq &\mathcal{L}(\mathcal{Z}^{(t+1)}, \{U_k^{(t+1)}\}_{k=s+1}^h, \mathcal{E}^{(t+1)}, \mathcal{Y}^{(t)}, \mu^{(t)}) + \frac{\eta^{(t)}}{2}\|\mathcal{Z}^{(t+1)} - \mathcal{Z}^{(t)}\|_F^2 \\
&+ \frac{\eta^{(t)}}{2} \sum_{k=s+1}^h \|U_k^{(t+1)} - U_k^{(t)}\|_F^2 + \frac{\eta^{(t)}}{2}\|\mathcal{E}^{(t+1)} - \mathcal{E}^{(t)}\|_F^2 \\
\leq &\mathcal{L}(\mathcal{Z}^{(t)}, \{U_k^{(t+1)}\}_{k=s+1}^h, \mathcal{E}^{(t+1)}, \mathcal{Y}^{(t)}, \mu^{(t)}) + \frac{\eta^{(t)}}{2} \sum_{k=s+1}^h \|U_k^{(t+1)} - U_k^{(t)}\|_F^2 \\
&+ \frac{\eta^{(t)}}{2}\|\mathcal{E}^{(t+1)} - \mathcal{E}^{(t)}\|_F^2 \\
\leq &\mathcal{L}(\mathcal{Z}^{(t)}, \{U_k^{(t)}\}_{k=s+1}^h, \mathcal{E}^{(t+1)}, \mathcal{Y}^{(t)}, \mu^{(t)}) + \frac{\eta^{(t)}}{2}\|\mathcal{E}^{(t+1)} - \mathcal{E}^{(t)}\|_F^2 \\
\leq &\mathcal{L}(\mathcal{Z}^{(t)}, \{U_k^{(t)}\}_{k=s+1}^h, \mathcal{E}^{(t)}, \mathcal{Y}^{(t)}, \mu^{(t)}) \\
= &\mathcal{L}(\mathcal{Z}^{(t)}, \{U_k^{(t)}\}_{k=s+1}^h, \mathcal{E}^{(t)}, \mathcal{Y}^{(t-1)}, \mu^{(t-1)}) + \frac{1}{2}(\mu^{(t-1)})^{-2}(\mu^{(t-1)} + \mu^{(t)})\|\mathcal{Y}^{(t)} - \mathcal{Y}^{(t-1)}\|_F^2.
\end{aligned}
$$
(7)

Therefore, we have

$$
\begin{aligned}
\|\mathcal{Z}^{(t+1)}\|_{\mathcal{U},1} \leq &\mathcal{L}(\mathcal{Z}^{(t+1)}, \{U_k^{(t+1)}\}_{k=s+1}^h, \mathcal{E}^{(t+1)}, \mathcal{Y}^{(t)}, \mu^{(t)}) + \|\mathcal{Y}^{(t)}\|_F^2/(\mu^{(t)})^2 \\
\leq &\mathcal{L}(\mathcal{Z}^{(t)}, \{U_k^{(t)}\}_{k=s+1}^h, \mathcal{E}^{(t)}, \mathcal{Y}^{(t)}, \mu^{(t)}) + \|\mathcal{Y}^{(t)}\|_F^2/(\mu^{(t)})^2 \\
\leq &\mathcal{L}(\mathcal{Z}^{(1)}, \{U_k^{(1)}\}_{k=s+1}^h, \mathcal{E}^{(1)}, \mathcal{Y}^{(0)}, \mu^{(0)}) \\
&+ \frac{1}{2}\sum_{n=1}^t (\mu^{(n-1)})^{-2}(\mu^{(n-1)} + \mu^{(n)})\|\mathcal{Y}^{(n)} - \mathcal{Y}^{(n-1)}\|_F^2 + \|\mathcal{Y}^{(t)}\|_F^2/(\mu^{(t)})^2 \\
\leq &\mathcal{L}(\mathcal{Z}^{(1)}, \{U_k^{(1)}\}_{k=s+1}^h, \mathcal{E}^{(1)}, \mathcal{Y}^{(0)}, \mu^{(0)}) + \sum_{n=1}^t (\mu^{(n-1)})^{-2}\mu^{(n)}\|\mathcal{Y}^{(n)} - \mathcal{Y}^{(n-1)}\|_F^2 \\
&+ \|\mathcal{Y}^{(t)}\|_F^2/(\mu^{(t)})^2.
\end{aligned}
$$
(8)

From (8), $\sum_{t=1}^\infty (\mu^{(t)})^{-2}\mu^{(t+1)} < +\infty$, and the boundedness of $\mathcal{Y}^{(t)}$, we can know that $\mathcal{Z}^{(t)}$ is bounded. Besides, since $\|U_k^{(t)}\|_F = \sqrt{I_k}$ holds for any positive integer $t$, $U_k^{(t)}$ and $\mathcal{X}^{(t)}$ are bounded. Therefore, $\mathcal{E}^{(t)}$ is bounded from $\lim_{t \longrightarrow \infty} \Psi_{\mathbb{I}}(\mathcal{M}) - \mathcal{X}^{(t+1)} - \mathcal{E}^{(t+1)} = \mathbf{0}$.

(iii) From (7), we have

$$
\begin{aligned}
&\sum_{t=1}^n \frac{\eta^{(t)}}{2}(\|\mathcal{Z}^{(t+1)} - \mathcal{Z}^{(t)}\|_F^2 + \sum_{k=s+1}^h \|U_k^{(t)} - U_k^{(t+1)}\|_F^2 + \|\mathcal{E}^{(t+1)} - \mathcal{E}^{(t)}\|_F^2) \\
&- \sum_{t=1}^n \frac{1}{2}(\mu^{(t-1)})^{-2}(\mu^{(t-1)} + \mu^{(t)})\|\mathcal{Y}^{(t)} - \mathcal{Y}^{(t-1)}\|_F^2 \\
\leq &\mathcal{L}(\mathcal{Z}^{(1)}, \{U_k^{(1)}\}_{k=s+1}^h, \mathcal{E}^{(1)}, \mathcal{Y}^{(0)}, \mu^{(0)}) - \mathcal{L}(\mathcal{Z}^{(n+1)}, \{U_k^{(n+1)}\}_{k=s+1}^h, \mathcal{E}^{(n+1)}, \mathcal{Y}^{(n)}, \mu^{(n)}) \\
\leq &\mathcal{L}(\mathcal{Z}^{(1)}, \{U_k^{(1)}\}_{k=s+1}^h, \mathcal{E}^{(1)}, \mathcal{Y}^{(0)}, \mu^{(0)}) + \|\mathcal{Y}^{(n)}\|_F^2/(\mu^{(n)})^2
\end{aligned}
$$
(9)

Since $\boldsymbol{\mathcal{Y}}^{(n)}$ is bounded, there exists $M_0$ and $M_1$ such that

$$\sum_{t=1}^{n} \frac{\eta^{(t)}}{2}(\|\boldsymbol{\mathcal{Z}}^{(t+1)} - \boldsymbol{\mathcal{Z}}^{(t)}\|_F^2 + \sum_{k=s+1}^{h} \|\boldsymbol{U}_k^{(t+1)} - \boldsymbol{U}_k^{(t)}\|_F^2 + \|\boldsymbol{\mathcal{E}}^{(t+1)} - \boldsymbol{\mathcal{E}}^{(t)}\|_F^2)$$

$$\leq M_0 + \sum_{t=1}^{n} \frac{1}{2}(\mu^{(t-1)})^{-2}(\mu^{(t-1)} + \mu^{(t)})M_1 \leq M_0 + \sum_{t=1}^{n}(\mu^{(t-1)})^{-2}\mu^{(t)}M_1. \tag{10}$$

As $n$ approaches infinity, we have

$$\sum_{t=1}^{\infty} \frac{\eta^{(t)}}{2}(\|\boldsymbol{\mathcal{Z}}^{(t+1)} - \boldsymbol{\mathcal{Z}}^{(t)}\|_F^2 + \sum_{k=s+1}^{h} \|\boldsymbol{U}_k^{(t+1)} - \boldsymbol{U}_k^{(t)}\|_F^2 + \|\boldsymbol{\mathcal{E}}^{(t+1)} - \boldsymbol{\mathcal{E}}^{(t)}\|_F^2)$$

$$\leq M_0 + \sum_{t=1}^{\infty}(\mu^{(t-1)})^{-2}\mu^{(t)}M_1 < \infty. \tag{11}$$

(iv) From (iii) we can see that there exists $M_2$ such that

$$\max(\|\boldsymbol{\mathcal{Z}}^{(t+1)} - \boldsymbol{\mathcal{Z}}^{(t)}\|_F^2, \|\boldsymbol{\mathcal{E}}^{(t+1)} - \boldsymbol{\mathcal{E}}^{(t)}\|_F^2, \{\|\boldsymbol{U}_k^{(t+1)} - \boldsymbol{U}_k^{(t)}\|_F^2\}_{k=s+1}^{h}) \leq (\eta^{(t)})^{(-1)}M_2^2,$$

therefore

$$\|\boldsymbol{\mathcal{X}}^{(t+1)} - \boldsymbol{\mathcal{X}}^{(t)}\|_F$$
$$=\|\boldsymbol{\mathcal{Z}}^{(t+1)} \times_h (\boldsymbol{U}_h^{(t+1)})^T \cdots \times_{s+1} (\boldsymbol{U}_{s+1}^{(t+1)})^T - \boldsymbol{\mathcal{Z}}^{(t)} \times_h (\boldsymbol{U}_h^{(t+1)})^T \cdots \times_{s+1} (\boldsymbol{U}_{s+1}^{(t+1)})^T$$
$$\quad + \boldsymbol{\mathcal{Z}}^{(t)} \times_h (\boldsymbol{U}_h^{(t+1)})^T \cdots \times_{s+1} (\boldsymbol{U}_{s+1}^{(t+1)})^T - \boldsymbol{\mathcal{Z}}^{(t)} \times_h (\boldsymbol{U}_h^{(t)})^T \cdots \times_{s+1} (\boldsymbol{U}_{s+1}^{(t)})^T\|_F$$
$$\leq\|\boldsymbol{\mathcal{Z}}^{(t+1)} - \boldsymbol{\mathcal{Z}}^{(t)}\|_F + \|\boldsymbol{\mathcal{Z}}^{(t)} \times_h (\boldsymbol{U}_h^{(t+1)})^T \cdots \times_{s+1} (\boldsymbol{U}_{s+1}^{(t+1)})^T - \boldsymbol{\mathcal{Z}}^{(t)} \times_h (\boldsymbol{U}_h^{(t)})^T \cdots \times_{s+1} (\boldsymbol{U}_{s+1}^{(t)})^T\|_F$$
$$=\|\boldsymbol{\mathcal{Z}}^{(t+1)} - \boldsymbol{\mathcal{Z}}^{(t)}\|_F + \|\boldsymbol{\mathcal{Z}}^{(t)} \times_h (\boldsymbol{U}_h^{(t+1)})^T \cdots \times_{s+1} (\boldsymbol{U}_{s+1}^{(t+1)})^T - \boldsymbol{\mathcal{Z}}^{(t)} \times_h (\boldsymbol{U}_h^{(t+1)})^T \cdots \times_{s+1} (\boldsymbol{U}_{s+1}^{(t)})^T$$
$$\quad + \boldsymbol{\mathcal{Z}}^{(t)} \times_h (\boldsymbol{U}_h^{(t+1)})^T \cdots \times_{s+1} (\boldsymbol{U}_{s+1}^{(t)})^T - \boldsymbol{\mathcal{Z}}^{(t)} \times_h (\boldsymbol{U}_h^{(t)})^T \cdots \times_{s+1} (\boldsymbol{U}_{s+1}^{(t)})^T\|_F$$
$$\leq\|\boldsymbol{\mathcal{Z}}^{(t+1)} - \boldsymbol{\mathcal{Z}}^{(t)}\|_F + \|\boldsymbol{\mathcal{Z}}^{(t)} \times_h (\boldsymbol{U}_h^{(t+1)})^T \cdots \times_{s+1} (\boldsymbol{U}_{s+1}^{(t+1)})^T - \boldsymbol{\mathcal{Z}}^{(t)} \times_h (\boldsymbol{U}_h^{(t+1)})^T \cdots \times_{s+1} (\boldsymbol{U}_{s+1}^{(t)})^T\|_F$$
$$\quad + \|\boldsymbol{\mathcal{Z}}^{(t)} \times_h (\boldsymbol{U}_h^{(t+1)})^T \cdots \times_{s+1} (\boldsymbol{U}_{s+1}^{(t)})^T - \boldsymbol{\mathcal{Z}}^{(t)} \times_h (\boldsymbol{U}_h^{(t)})^T \cdots \times_{s+1} (\boldsymbol{U}_{s+1}^{(t)})^T\|_F$$
$$\leq\|\boldsymbol{\mathcal{Z}}^{(t+1)} - \boldsymbol{\mathcal{Z}}^{(t)}\|_F + \|\boldsymbol{U}_{s+1}^{(t+1)} - \boldsymbol{U}_{s+1}^{(t)}\|_F\|\boldsymbol{\mathcal{Z}}^{(t)}\|_F + \|\boldsymbol{\mathcal{Z}}^{(t)} \times_h (\boldsymbol{U}_h^{(t+1)})^T \cdots \times_{s+2} (\boldsymbol{U}_{s+2}^{(t+1)})^T$$
$$\quad - \boldsymbol{\mathcal{Z}}^{(t)} \times_h (\boldsymbol{U}_h^{(t)})^T \cdots \times_{s+2} (\boldsymbol{U}_{s+2}^{(t)})^T\|_F$$
$$\leq\|\boldsymbol{\mathcal{Z}}^{(t+1)} - \boldsymbol{\mathcal{Z}}^{(t)}\|_F + \sum_{k=s+1}^{h} (\|\boldsymbol{U}_k^{(t+1)} - \boldsymbol{U}_k^{(t)}\|_F)\|\boldsymbol{\mathcal{Z}}^{(t)}\|_F$$
$$\leq(\eta^{(t)})^{(-1/2)}(1 + h\|\boldsymbol{\mathcal{Z}}^{(t)}\|_F)M_2. \tag{12}$$

From the boundedness of $\boldsymbol{\mathcal{Z}}^{(t)}$, there exists $M_3$ such that $\|\boldsymbol{\mathcal{X}}^{(t+1)} - \boldsymbol{\mathcal{X}}^{(t)}\|_F^2 \leq (\eta^{(t)})^{(-1)}M_3$.

Let $\boldsymbol{\mathcal{D}}^{(t+1)} = \Psi_{\mathbb{I}}(\boldsymbol{\mathcal{M}}) - \boldsymbol{\mathcal{X}}^{(t+1)} - \boldsymbol{\mathcal{E}}^{(t+1)}$. From the above discussion, we know that there exists $M_4$ such that $\|\boldsymbol{\mathcal{D}}^{(t+1)} - \boldsymbol{\mathcal{D}}^{(t)}\|_F \leq \|\boldsymbol{\mathcal{X}}^{(t+1)} - \boldsymbol{\mathcal{X}}^{(t)}\|_F + \|\boldsymbol{\mathcal{E}}^{(t+1)} - \boldsymbol{\mathcal{E}}^{(t)}\|_F \leq (\eta^{(t)})^{(-1/2)}M_4$. Thus, we have

$$\|\boldsymbol{\mathcal{D}}^{(t)}\|_F \leq (\eta^{(t)})^{(-1/2)}M_4 + \|\boldsymbol{\mathcal{D}}^{(t+1)}\|_F \leq M_4 \sum_{n=0}^{m}(\eta^{(t+n)})^{(-1/2)} + \|\boldsymbol{\mathcal{D}}^{(t+1+m)}\|_F$$

for any $m > 0$ and $(\mu^{(n)})^{(-1)}\|\boldsymbol{\mathcal{Y}}^{(n+1)} - \boldsymbol{\mathcal{Y}}^{(n)}\|_F = \|\boldsymbol{\mathcal{D}}^{(n+1)}\|_F \leq M_4 \sum_{t=n+1}^{+\infty}(\eta^{(t)})^{(-1/2)}$ when $m \longrightarrow \infty$.

From $\lim\limits_{n \longrightarrow \infty} \mu^{(n)} \sum\limits_{t=n}^{\infty}(\eta^{(t)})^{-1/2} = 0$, we have $\lim\limits_{n \longrightarrow \infty} \|\boldsymbol{\mathcal{Y}}^{(n+1)} - \boldsymbol{\mathcal{Y}}^{(n)}\|_F = 0$.

(v) From the boundedness of $\{[\boldsymbol{\mathcal{Z}}^{(t)}, \{\boldsymbol{U}_k^{(t)}\}_{k=s+1}^{h}, \boldsymbol{\mathcal{X}}^{(t)}, \boldsymbol{\mathcal{E}}^{(t)}]\}$, there exist a subsequence $\{[\boldsymbol{\mathcal{Z}}^{(t_i)}, \{\boldsymbol{U}_k^{(t_i)}\}_{k=s+1}^{h}, \boldsymbol{\mathcal{E}}^{(t_i)}, \boldsymbol{\mathcal{Y}}^{(t_i)}]\}$ and $[\boldsymbol{\mathcal{Z}}^*, \{\boldsymbol{U}_k^*\}_{k=s+1}^{h}, \boldsymbol{\mathcal{E}}^*, \boldsymbol{\mathcal{Y}}^*]$ such that

$\lim_{i \to +\infty} [\boldsymbol{\mathcal{Z}}^{(t_i)}, \{\boldsymbol{U}_k^{(t_i)}\}_{k=s+1}^h, \boldsymbol{\mathcal{E}}^{(t_i)}, \boldsymbol{\mathcal{Y}}^{(t_i)}] = [\boldsymbol{\mathcal{Z}}^*, \{\boldsymbol{U}_k^*\}_{k=s+1}^h, \boldsymbol{\mathcal{E}}^*, \boldsymbol{\mathcal{Y}}^*]$. From the optimality of $\boldsymbol{\mathcal{Z}}^{(t_i+1)}$ and the convexity of the tensor $U_1$ norm, there exists $\boldsymbol{\mathcal{H}}^{(t_i+1)} \in \partial \|\boldsymbol{\mathcal{Z}}^{(t_i+1)}\|_{\mathcal{U},1}$ such that

$$\boldsymbol{\mathcal{H}}^{(t_i+1)} + \mu^{(t_i)}(\boldsymbol{\mathcal{Z}}^{(t_i+1)} - \boldsymbol{\mathcal{P}}^{(t_i)} \times_{s+1} \boldsymbol{U}_{s+1}^{(t_i)} \times_{s+2} \cdots \times_h \boldsymbol{U}_h^{(t_i)}) + \eta^{(t_i)}(\boldsymbol{\mathcal{Z}}^{(t_i+1)} - \boldsymbol{\mathcal{Z}}^{(t_i)}) = \boldsymbol{0}$$

and

$$\boldsymbol{\mathcal{H}}^* - \boldsymbol{\mathcal{Y}}^* \times_{s+1} \boldsymbol{U}_{s+1}^* \times_{s+2} \cdots \times_h \boldsymbol{U}_h^* = \boldsymbol{0},$$

where $\lim_{i \to +\infty} \boldsymbol{\mathcal{H}}^{(t_i+1)} = \boldsymbol{\mathcal{H}}^*$, and $\hat{\boldsymbol{\mathcal{P}}}^{(t_i)} = \Psi(\boldsymbol{\mathcal{M}}) - \boldsymbol{\mathcal{E}}^{(t_i)} + \frac{1}{\mu^{(t_i)}}\boldsymbol{\mathcal{Y}}^{(t_i)}$. By the upper semi-continuous property of the subdifferential Clarke (1983), $\boldsymbol{\mathcal{Y}}^* \times_{s+1} \boldsymbol{U}_{s+1}^* \times_{s+2} \cdots \times_h \boldsymbol{U}_h^* = \boldsymbol{\mathcal{H}}^* \in \partial \|\boldsymbol{\mathcal{Z}}^*\|_{\mathcal{U},1}$.

From the optimality of $\boldsymbol{U}_k^{(t_i+1)}$, we have $(\boldsymbol{U}_k^{(t_i+1)})^T \boldsymbol{U}_k^{(t_i+1)} = \boldsymbol{I}$, and there exists $\boldsymbol{Y}_k^{(t_i+1)}$ such that $\boldsymbol{0} = \mu^{(t_i)}(\boldsymbol{U}_k^{(t_i+1)}\boldsymbol{\mathcal{B}}_{(k)} - \boldsymbol{\mathcal{A}}_{(k)})\boldsymbol{\mathcal{B}}_{(k)}^T + \eta^{(t_i)}(\boldsymbol{U}_k^{(t_i+1)} - \boldsymbol{U}_k^{(t_i)}) + \boldsymbol{U}_k^{(t_i+1)}(\boldsymbol{Y}_k^{(t_i+1)} + (\boldsymbol{Y}_k^{(t_i+1)})^T)$, where $\boldsymbol{\mathcal{B}} = \hat{\boldsymbol{\mathcal{P}}}^{(t_i)} \times_{s+1} \boldsymbol{U}_{s+1}^{(t_i+1)} \cdots \times_{k-1} \boldsymbol{U}_{k-1}^{(t_i+1)}$ and $\boldsymbol{\mathcal{A}} = \boldsymbol{\mathcal{Z}}^{(t_i+1)} \times_h \boldsymbol{U}_h^{(t_i)T} \times_{h-1} \cdots \times_{k+1} \boldsymbol{U}_{k+1}^{(t_i)T}$.

Thus, we have $(\boldsymbol{U}_k^*)^T \boldsymbol{U}_k^* = \boldsymbol{I}$ and there exists $\boldsymbol{Y}_k^*$ such that $\boldsymbol{0} = (\boldsymbol{U}_k^* \boldsymbol{\mathcal{C}}_{(k)}^*)\boldsymbol{\mathcal{B}}_{(k)}^{*T} + \boldsymbol{U}_k^*(\boldsymbol{Y}_k^* + (\boldsymbol{Y}_k^*)^T)$ if $i \longrightarrow \infty$, where $\boldsymbol{\mathcal{B}}^* = \boldsymbol{\mathcal{Z}}^* \times_h (\boldsymbol{U}_h^*)^T \cdots \times_k (\boldsymbol{U}_k^*)^T$ and $\boldsymbol{\mathcal{C}}^* = \boldsymbol{\mathcal{Y}}^* \times_{s+1} \boldsymbol{U}_{s+1}^* \cdots \times_{k-1} \boldsymbol{U}_{k-1}^*$. Therefore, $\boldsymbol{0} = -\boldsymbol{\mathcal{F}}_{(k)}^*(\boldsymbol{\mathcal{C}}_{(k)}^*)^T + \boldsymbol{U}_k^*(-\boldsymbol{Y}_k^* + (-\boldsymbol{Y}_k^*)^T)$ holds, where $\boldsymbol{\mathcal{F}}^* = \boldsymbol{\mathcal{Z}}^* \times_h (\boldsymbol{U}_h^*)^T \cdots \times_{k+1} (\boldsymbol{U}_{k+1}^*)^T$.

Besides, from the optimality of $\boldsymbol{\mathcal{E}}^{(t_i+1)}$, we have $\Psi_{\mathbb{I}}(\boldsymbol{\mathcal{E}}^{(t_i+1)}) = \boldsymbol{0}$ and

$$\Psi_{\mathbb{I}^c}(\mu^{(t_i)}(\boldsymbol{\mathcal{E}}^{(t_i+1)} + \boldsymbol{\mathcal{X}}^{(t_i+1)} - \frac{1}{\mu^{(t_i)}}\boldsymbol{\mathcal{Y}}^{(t_i)}) + \eta^{(t_i)}(\boldsymbol{\mathcal{E}}^{(t_i+1)} - \boldsymbol{\mathcal{E}}^{(t_i)})) = \boldsymbol{0},$$

from which we deduce that both of $\Psi_{\mathbb{I}}(\boldsymbol{\mathcal{E}}^*) = \boldsymbol{0}$ and $\boldsymbol{0} = \lim_{i \longrightarrow \infty} \Psi_{\mathbb{I}^c}(\mu^{(t_i)}(\boldsymbol{\mathcal{E}}^{(t_i+1)} - \Psi_{\mathbb{I}}(\boldsymbol{\mathcal{M}}) + \boldsymbol{\mathcal{X}}^{(t_i+1)} - \frac{1}{\mu^{(t_i)}}\boldsymbol{\mathcal{Y}}^{(t_i)}) = -\Psi_{\mathbb{I}^c}(\boldsymbol{\mathcal{Y}}^*)$ hold. Furthermore, it is evident that there exists $\boldsymbol{\mathcal{W}}^*$ such that $\boldsymbol{0} = -\Psi_{\mathbb{I}}(\boldsymbol{\mathcal{Y}}^*) + \boldsymbol{\mathcal{W}}^*$. $\qquad\square$

## 5 PROOF OF LEMMA 1 AND THEOREM 2

**Lemma 1.** *For $\boldsymbol{\mathcal{A}} \in \mathbb{R}^{I_1 \times I_2 \times \cdots \times I_h}$, the subgradient of $\|\boldsymbol{\mathcal{A}}\|_{1,\mathcal{U}}$ is given as $\partial \|\boldsymbol{\mathcal{A}}\|_{1,\mathcal{U}} = \{\mathcal{U}^{-1}(\mathrm{sgn}(\mathcal{U}(\boldsymbol{\mathcal{A}})) + \boldsymbol{\mathcal{F}}|\Psi_{\hat{\mathbb{H}}}(\mathcal{U}(\boldsymbol{\mathcal{F}})) = \boldsymbol{0}, \|\boldsymbol{\mathcal{F}}\|_{\mathcal{U},\infty} \le 1\}$, where $\hat{\mathbb{H}}$ denotes the support of $\mathcal{U}(\boldsymbol{\mathcal{A}})$.*

*Proof.* **We can get the conclusion by** $\langle \mathcal{U}^{-1}(\mathrm{sgn}(\mathcal{U}(\boldsymbol{\mathcal{A}}))) + \boldsymbol{\mathcal{F}}, \boldsymbol{\mathcal{A}} \rangle = \langle \mathrm{sgn}(\mathcal{U}(\boldsymbol{\mathcal{A}})), \mathcal{U}(\boldsymbol{\mathcal{A}}) \rangle + \langle \mathcal{U}(\boldsymbol{\mathcal{F}}), \mathcal{U}(\boldsymbol{\mathcal{A}}) \rangle = \|\boldsymbol{\mathcal{A}}\|_{1,\mathcal{U}}$ **and** $\|\mathcal{U}^{-1}(\mathrm{sgn}(\mathcal{U}(\boldsymbol{\mathcal{A}})) + \boldsymbol{\mathcal{F}}\|_{\mathcal{U},\infty} = \|\mathrm{sgn}(\mathcal{U}(\boldsymbol{\mathcal{A}})) + \mathcal{U}(\boldsymbol{\mathcal{F}})\|_{\infty} = \max(\|\mathrm{sgn}(\mathcal{U}(\boldsymbol{\mathcal{A}}))\|_{\infty}), \|\mathcal{U}(\boldsymbol{\mathcal{F}})\|_{\infty}) \le 1$ **Watson (1992).** $\qquad\square$

**Lemma 2.** *If there exists a dual certificate $\boldsymbol{\mathcal{G}}$ (that satisfy $\Psi_{\mathbb{I}}(\boldsymbol{\mathcal{G}}) = \boldsymbol{\mathcal{G}}$, $P_{\mathbb{S}}(\boldsymbol{\mathcal{G}}) = \hat{\mathcal{U}}^{-1}(\mathrm{sgn}(\hat{\mathcal{U}}(\boldsymbol{\mathcal{M}})))$ and $\|P_{\mathbb{S}^\perp}(\boldsymbol{\mathcal{G}})\|_{\hat{\mathcal{U}},\infty} \le 1$ ) and any $\boldsymbol{\mathcal{H}}$ obeying $\Psi_{\mathbb{I}}(\boldsymbol{\mathcal{H}}) = \boldsymbol{0}$, then*

$$\|\boldsymbol{\mathcal{M}} + \boldsymbol{\mathcal{H}}\|_{\hat{\mathcal{U}},1} \ge \|\boldsymbol{\mathcal{M}}\|_{\hat{\mathcal{U}},1} + (1 - \|P_{\mathbb{S}^\perp}(\boldsymbol{\mathcal{G}})\|_{\hat{\mathcal{U}},\infty})\|P_{\mathbb{S}^\perp}(\boldsymbol{\mathcal{H}})\|_{\hat{\mathcal{U}},1}.$$

*Proof.* **For any** $\boldsymbol{\mathcal{Z}} \in \partial \|\boldsymbol{\mathcal{M}}\|_{\hat{\mathcal{U}},1}$, **we have** $\|\boldsymbol{\mathcal{M}} + \boldsymbol{\mathcal{H}}\|_{\hat{\mathcal{U}},1} \ge \|\boldsymbol{\mathcal{M}}\|_{\hat{\mathcal{U}},1} + \langle \boldsymbol{\mathcal{Z}}, \boldsymbol{\mathcal{H}} \rangle$. **Since** $\boldsymbol{\mathcal{G}} = \hat{\mathcal{U}}^{-1}(\mathrm{sgn}(\hat{\mathcal{U}}(\boldsymbol{\mathcal{M}}))) + P_{\mathbb{S}^\perp}(\boldsymbol{\mathcal{G}})$ **and** $\boldsymbol{\mathcal{Z}} = \hat{\mathcal{U}}^{-1}(\mathrm{sgn}(\hat{\mathcal{U}}(\boldsymbol{\mathcal{M}}))) + P_{\mathbb{S}^\perp}(\boldsymbol{\mathcal{Z}})$, **we obtain** $\|\boldsymbol{\mathcal{M}} + \boldsymbol{\mathcal{H}}\|_{\hat{\mathcal{U}},1} \ge \|\boldsymbol{\mathcal{M}}\|_{\hat{\mathcal{U}},1} + \langle \boldsymbol{\mathcal{G}}, \boldsymbol{\mathcal{H}} \rangle + \langle P_{\mathbb{S}^\perp}(\boldsymbol{\mathcal{Z}} - \boldsymbol{\mathcal{G}}), \boldsymbol{\mathcal{H}} \rangle$. **Therefore we have** $\|\boldsymbol{\mathcal{M}} + \boldsymbol{\mathcal{H}}\|_{\hat{\mathcal{U}},1} \ge \|\boldsymbol{\mathcal{M}}\|_{\hat{\mathcal{U}},1} + \langle P_{\mathbb{S}^\perp}(\boldsymbol{\mathcal{Z}} - \boldsymbol{\mathcal{G}}), \boldsymbol{\mathcal{H}} \rangle$, **where** $\langle \boldsymbol{\mathcal{G}}, \boldsymbol{\mathcal{H}} \rangle = 0$ **due to** $\Psi_{\mathbb{I}}(\boldsymbol{\mathcal{H}}) = \boldsymbol{0}$.

**Since** $\|\cdot\|_{\hat{\mathcal{U}},1}$ **and** $\|\cdot\|_{\hat{\mathcal{U}},\infty}$ **are dual to each other, there exists** $\|\boldsymbol{\mathcal{Z}}_0\|_{\hat{\mathcal{U}},\infty} \le 1$ **such that** $\langle \boldsymbol{\mathcal{Z}}_0, P_{\mathbb{S}^\perp}(\boldsymbol{\mathcal{H}}) \rangle = \|P_{\mathbb{S}^\perp}(\boldsymbol{\mathcal{H}})\|_{\hat{\mathcal{U}},1}$. **Hence, by selecting a** $\boldsymbol{\mathcal{Z}}$ **such that** $P_{\mathbb{S}^\perp}(\boldsymbol{\mathcal{Z}}) = P_{\mathbb{S}^\perp}(\boldsymbol{\mathcal{Z}}_0)$, **we get** $\langle P_{\mathbb{S}^\perp}(\boldsymbol{\mathcal{Z}}), \boldsymbol{\mathcal{H}} \rangle = \|P_{\mathbb{S}^\perp}(\boldsymbol{\mathcal{H}})\|_{\hat{\mathcal{U}},1}$. **Therefore, we have** $\langle P_{\mathbb{S}^\perp}(\boldsymbol{\mathcal{Z}} - \boldsymbol{\mathcal{G}}), \boldsymbol{\mathcal{H}} \rangle \ge (1 - \|P_{\mathbb{S}^\perp}(\boldsymbol{\mathcal{G}})\|_{\hat{\mathcal{U}},\infty})\|P_{\mathbb{S}^\perp}(\boldsymbol{\mathcal{H}})\|_{\hat{\mathcal{U}},1}$ **due to** $|\langle P_{\mathbb{S}^\perp}(\boldsymbol{\mathcal{G}}), P_{\mathbb{S}^\perp}(\boldsymbol{\mathcal{H}}) \rangle| \le \|P_{\mathbb{S}^\perp}(\boldsymbol{\mathcal{G}})\|_{\hat{\mathcal{U}},\infty}\|P_{\mathbb{S}^\perp}(\boldsymbol{\mathcal{H}})\|_{\hat{\mathcal{U}},1}$, **thus completed the proof.** $\qquad\square$

$$\min_{\boldsymbol{\mathcal{X}}, \boldsymbol{U}_{k_n}^T \boldsymbol{U}_{k_n} = \boldsymbol{I}(n=s+1,\cdots,h)} \|\boldsymbol{\mathcal{X}} \times_{k_{s+1}} \boldsymbol{U}_{k_{s+1}} \cdots \times_{k_h} \boldsymbol{U}_{k_h}\|_{\mathcal{U},1}$$

$$s.t. \ \|\Psi_{\mathbb{I}}(\boldsymbol{\mathcal{M}}) - \Psi_{\mathbb{I}}(\boldsymbol{\mathcal{X}})\|_F \leq \delta \quad (13)$$

**Theorem 2.** *If the dual certificate* $\boldsymbol{\mathcal{G}} = \Psi_{\mathbb{I}} P_{\mathbb{S}} (P_{\mathbb{S}} \Psi_{\mathbb{I}} P_{\mathbb{S}})^{-1} (\hat{\mathcal{U}}^{-1}(\mathrm{sgn}(\hat{\mathcal{U}}(\boldsymbol{\mathcal{M}}))))$ *satisfies* $\|P_{\mathbb{S}^\perp}(\boldsymbol{\mathcal{G}})\|_{\hat{\mathcal{U}},\infty} \leq C_1 < 1$ *and* $P_{\mathbb{S}} \Psi_{\mathbb{I}} P_{\mathbb{S}} \succcurlyeq C_2 p \mathcal{I}$, *then we can obtain the following inequality:*

$$\|\boldsymbol{\mathcal{M}} - \hat{\boldsymbol{\mathcal{X}}}\|_F \leq \frac{1}{1-C_1} \sqrt{\frac{1/C_2+p}{p} I_1 I_2} \delta + \delta, \quad (14)$$

*where* $\hat{\boldsymbol{\mathcal{X}}}$ *is obtained by* (13) *and* $p$ *denotes the sampling rate.*

*Proof.* Let $\boldsymbol{\mathcal{H}}$ be $\boldsymbol{\mathcal{H}} = \hat{\boldsymbol{\mathcal{X}}} - \boldsymbol{\mathcal{M}}$ for brevity. Considering that $\|\boldsymbol{\mathcal{H}}\|_F = \|\Psi_{\mathbb{I}}(\boldsymbol{\mathcal{H}})\|_F + \|\Psi_{\mathbb{I}^c}(\boldsymbol{\mathcal{H}})\|_F \leq \delta + \|\Psi_{\mathbb{I}^c}(\boldsymbol{\mathcal{H}})\|_F$, **we focus solely on the second term** $\|\Psi_{\mathbb{I}^c}(\boldsymbol{\mathcal{H}})\|_F$ **in the following discussion.**

**Utilizing the triangle inequality and Lemma 2, we obtain** $\|\boldsymbol{\mathcal{M}} + \boldsymbol{\mathcal{H}}\|_{\hat{\mathcal{U}},1} \geq \|\boldsymbol{\mathcal{M}} + \Psi_{\mathbb{I}^c}(\boldsymbol{\mathcal{H}})\|_{\hat{\mathcal{U}},1} - \|\Psi_{\mathbb{I}}(\boldsymbol{\mathcal{H}})\|_{\hat{\mathcal{U}},1}$ **and** $\|\boldsymbol{\mathcal{M}} + \Psi_{\mathbb{I}^c}(\boldsymbol{\mathcal{H}})\|_{\hat{\mathcal{U}},1} \geq \|\boldsymbol{\mathcal{M}}\|_{\hat{\mathcal{U}},1} + (1 - \|P_{\mathbb{S}^\perp}(\boldsymbol{\mathcal{G}})\|_{\hat{\mathcal{U}},\infty}) \|P_{\mathbb{S}^\perp}(\Psi_{\mathbb{I}^c}(\boldsymbol{\mathcal{H}}))\|_{\hat{\mathcal{U}},1}$. **Consequently, we have** $\|\boldsymbol{\mathcal{M}}\|_{\hat{\mathcal{U}},1} \geq \|\boldsymbol{\mathcal{M}} + \boldsymbol{\mathcal{H}}\|_{\hat{\mathcal{U}},1} \geq \|\boldsymbol{\mathcal{M}}\|_{\hat{\mathcal{U}},1} + (1 - \|P_{\mathbb{S}^c}(\boldsymbol{\mathcal{G}})\|_{\hat{\mathcal{U}},\infty}) \|P_{\mathbb{S}^\perp}(\Psi_{\mathbb{I}^c}(\boldsymbol{\mathcal{H}}))\|_{\hat{\mathcal{U}},1} - \|\Psi_{\mathbb{I}}(\boldsymbol{\mathcal{H}})\|_{\hat{\mathcal{U}},1} \geq \|\boldsymbol{\mathcal{M}}\|_{\hat{\mathcal{U}},1} + (1 - C_1) \|P_{\mathbb{S}^\perp}(\Psi_{\mathbb{I}^c}(\boldsymbol{\mathcal{H}}))\|_{\hat{\mathcal{U}},1} - \|\Psi_{\mathbb{I}}(\boldsymbol{\mathcal{H}})\|_{\hat{\mathcal{U}},1}$, **which leads to** $\|P_{\mathbb{S}^\perp}(\Psi_{\mathbb{I}^c}(\boldsymbol{\mathcal{H}}))\|_F \leq \|P_{\mathbb{S}^\perp}(\Psi_{\mathbb{I}^c}(\boldsymbol{\mathcal{H}}))\|_{\hat{\mathcal{U}},1} \leq \frac{1}{1-C_1} \|\Psi_{\mathbb{I}}(\boldsymbol{\mathcal{H}})\|_{\hat{\mathcal{U}},1} \leq \frac{\sqrt{I_1 I_2}}{1-C_1} \|\Psi_{\mathbb{I}}(\boldsymbol{\mathcal{H}})\|_F \leq \frac{\sqrt{I_1 I_2}}{1-C_1} \delta.$

**Additionally, due to** $P_{\mathbb{S}} \Psi_{\mathbb{I}} P_{\mathbb{S}} \succcurlyeq C_2 p \mathcal{I}$, **we find** $\|\Psi_{\mathbb{I}}(P_{\mathbb{S}}(\Psi_{\mathbb{I}^c}(\boldsymbol{\mathcal{H}})))\|_F^2 = \langle P_{\mathbb{S}} \Psi_{\mathbb{I}} P_{\mathbb{S}}(\Psi_{\mathbb{I}^c}(\boldsymbol{\mathcal{H}})), P_{\mathbb{S}}(\Psi_{\mathbb{I}^c}(\boldsymbol{\mathcal{H}})) \rangle \geq C_2 p \|P_{\mathbb{S}}(\Psi_{\mathbb{I}^c}(\boldsymbol{\mathcal{H}}))\|_F^2$. **Moreover, because of** $\Psi_{\mathbb{I}}(P_{\mathbb{S}^\perp}(\Psi_{\mathbb{I}^c}(\boldsymbol{\mathcal{H}}))) + \Psi_{\mathbb{I}}(P_{\mathbb{S}}(\Psi_{\mathbb{I}^c}(\boldsymbol{\mathcal{H}}))) = \mathbf{0}$, **we get** $C_2 p \|P_{\mathbb{S}}(\Psi_{\mathbb{I}^c}(\boldsymbol{\mathcal{H}}))\|_F^2 \leq \|\Psi_{\mathbb{I}}(\Psi_{\mathbb{S}}(\Psi_{\mathbb{I}^c}(\boldsymbol{\mathcal{H}})))\|_F^2 = \|\Psi_{\mathbb{I}}(P_{\mathbb{S}^\perp}(\Psi_{\mathbb{I}^c}(\boldsymbol{\mathcal{H}})))\|_F^2 \leq \|P_{\mathbb{S}^\perp}(\Psi_{\mathbb{I}^c}(\boldsymbol{\mathcal{H}}))\|_F^2.$

**Consequently, we have** $\|\Psi_{\mathbb{I}^c}(\boldsymbol{\mathcal{H}})\|_F^2 = \|P_{\mathbb{S}^\perp}(\Psi_{\mathbb{I}^c}(\boldsymbol{\mathcal{H}}))\|_F^2 + \|P_{\mathbb{S}}(\Psi_{\mathbb{I}^c}(\boldsymbol{\mathcal{H}}))\|_F^2 \leq (\frac{1}{C_2 p} + 1) \|P_{\mathbb{S}^\perp}(\Psi_{\mathbb{I}^c}(\boldsymbol{\mathcal{H}}))\|_F^2 \leq (\frac{1}{C_2 p} + 1) \frac{I_1 I_2}{(1-C_1)^2} \delta^2$, **and thus completed the proof.** $\square$

## 6 STABLE TC-SL

**Similarly, we can establish stable TC-SL based on the given** $\{\hat{\boldsymbol{U}}_{k_n}\}_{n=3}^s$:

$$\min_{\boldsymbol{\mathcal{X}}, \boldsymbol{U}_{k_n}^T \boldsymbol{U}_{k_n} = \boldsymbol{I}(n=s+1,\cdots,h)} \|\boldsymbol{\mathcal{X}} \times_{k_{s+1}} \boldsymbol{U}_{k_{s+1}} \cdots \times_{k_h} \boldsymbol{U}_{k_h}\|_{*,\mathcal{U}}^{(k_1,k_2)}$$

$$s.t. \ \|\Psi_{\mathbb{I}}(\boldsymbol{\mathcal{M}}) - \Psi_{\mathbb{I}}(\boldsymbol{\mathcal{X}})\|_F \leq \delta. \quad (15)$$

**Before proving the stable recovery property of** (15), **we need to introduce the definition of the tensor product, which is a direct generalization from high order tensor product defined in Qin et al. (2022).**

**Definition 3.** *(tensor product for given* $(k_1, k_2)$ *and* $\mathcal{U}$*) For an h-order tensor* $\boldsymbol{\mathcal{A}} \in \mathbb{R}^{I_{k_1} \times L \times \cdots \times I_{k_h}}$ *and* $\boldsymbol{\mathcal{B}} \in \mathbb{R}^{L \times I_{k_2} \times \cdots \times I_{k_h}}$, *the tensor product of* $\boldsymbol{\mathcal{A}}$ *and* $\boldsymbol{\mathcal{B}}$ *is defined as* $\boldsymbol{\mathcal{A}} * \boldsymbol{\mathcal{B}} = \mathcal{U}^{-1}(\mathcal{U}(\boldsymbol{\mathcal{A}}) \odot_f \mathcal{U}(\boldsymbol{\mathcal{B}}))$, *where* $[\bar{\boldsymbol{\mathcal{A}}} \odot_f \bar{\boldsymbol{\mathcal{B}}}]_{:,:,i_{k_3},i_{k_4},\cdots,i_{k_h}} = [\bar{\boldsymbol{\mathcal{A}}}]_{:,:,i_{k_3},i_{k_4},\cdots,i_{k_h}} [\bar{\boldsymbol{\mathcal{B}}}]_{:,:,i_{k_3},i_{k_4},\cdots,i_{k_h}}.$

**Let** $\boldsymbol{\mathcal{A}} = \boldsymbol{\mathcal{U}} * \boldsymbol{\mathcal{S}} * \boldsymbol{\mathcal{V}}^T$ **be t-SVD of** $\boldsymbol{\mathcal{A}}$ **by using tensor product given in the Definition 3, where** $\boldsymbol{\mathcal{V}}^T$ **is defined by** $[\mathcal{U}(\boldsymbol{\mathcal{V}}^T)]_{:,:,i_{k_3},i_{k_4},\cdots,i_{k_h}} = [\mathcal{U}(\boldsymbol{\mathcal{V}})]_{:,:,i_{k_3},i_{k_4},\cdots,i_{k_h}}^T$ **for all** $(i_{k_3}, i_{k_4}, \cdots, i_{k_h})$. **For simplicity, we'll consider the case of** $(k_1, k_2, \cdots, k_h) = (1, 2, \cdots, h)$ **and use** $\|\cdot\|_{*,\mathcal{U}}$ **and** $\|\cdot\|_{2,\mathcal{U}}$ **to denote** $\|\cdot\|_{*,\mathcal{U}}^{(1,2)}$ **and** $\|\cdot\|_{2,\mathcal{U}}^{(1,2)}$, **respectively.**

**Lemma 3.** *For tensor* $\boldsymbol{\mathcal{A}} \in \mathbb{R}^{I_1 \times I_2 \times \cdots \times I_h}$ *with* $\mathrm{rank}_{(1,2)}(\mathcal{U}(\boldsymbol{\mathcal{A}})) = r$, *if its skinny t-SVD is* $\boldsymbol{\mathcal{A}} = \boldsymbol{\mathcal{U}} * \boldsymbol{\mathcal{S}} * \boldsymbol{\mathcal{V}}^T$, *then the subgradient of* $\|\boldsymbol{\mathcal{A}}\|_{*,\mathcal{U}}$ *can be given as* $\partial\|\boldsymbol{\mathcal{A}}\|_{*,\mathcal{U}} = \{\boldsymbol{\mathcal{U}} * \boldsymbol{\mathcal{V}}^T + \boldsymbol{\mathcal{W}} | \boldsymbol{\mathcal{U}}^T * \boldsymbol{\mathcal{W}} = 0, \boldsymbol{\mathcal{W}} * \boldsymbol{\mathcal{V}} = 0, \|\boldsymbol{\mathcal{W}}\|_{2,\mathcal{U}} \leq 1\}.$

Table 1: Comparing various methods on the five video segments at a sampling rate $p = 0.3$.

| Video | TNN-DCT | TNN-DFT | SNN | KBR | WSTNN | HTNN-DCT | TC-SL | TC-U1 |
|-------|---------|---------|-----|-----|-------|----------|-------|-------|
| *run* $9^{th}$ | 25.77 | 25.73 | 22.53 | 27.48 | 30.54 | 28.35 | 30.63 | **32.79** |
| *run* $39^{th}$ | 30.66 | 30.60 | 29.24 | 38.03 | 34.74 | 34.05 | 35.01 | **40.39** |
| *run* $40^{th}$ | 28.83 | 28.80 | 26.13 | 33.1 | 32.59 | 31.73 | 33.35 | **36.06** |
| *run* $42^{th}$ | 27.72 | 27.86 | 24.48 | 31.75 | 32.08 | 30.63 | 31.87 | **36.88** |
| *run* $108^{th}$ | 31.64 | 31.55 | 29.83 | 34.13 | 34.13 | 33.72 | 35.57 | **36.96** |
| Average | 28.92 | 28.91 | 26.44 | 32.90 | 32.82 | 31.70 | 33.29 | **36.62** |

*Proof.* **We can obtain the conclusion by** $\left\langle \mathcal{U} * \mathcal{V}^T + \mathcal{W}, \mathcal{A} \right\rangle = \left\langle \mathcal{U} * \mathcal{V}^T, \mathcal{U} * \mathcal{S} * \mathcal{V}^T \right\rangle + \left\langle \mathcal{W}, \mathcal{U} * \mathcal{S} * \mathcal{V}^T \right\rangle = \langle \mathcal{I}, \mathcal{S} \rangle = \|\mathcal{A}\|_{*,\mathcal{U}}$ **and** $\|\mathcal{U} * \mathcal{V}^T + \mathcal{W}\|_{2,\mathcal{U}} \leq 1$ **Watson (1992).** $\qquad\square$

**Suppose** $\mathcal{M} = \mathcal{U}_0 * \mathcal{S}_0 * \mathcal{V}_0^T$ **is the skinny t-SVD of** $\mathcal{M}$. **We define** $\mathbb{T} = \{\mathcal{U}_0 * \mathcal{Y}^T + \mathcal{W} * \mathcal{V}_0^T, \mathcal{Y}, \mathcal{W} \in \mathbb{R}^{I_1 \times r \times I_3 \times \cdots \times I_h}\}$, $P_{\mathbb{T}}$ **is the projections onto** $\mathbb{T}$, **and** $\mathbb{T}^\perp$ **is the orthogonal complement of** $\mathbb{T}$. **Considering** $(\hat{\mathcal{X}}, \{\hat{U}_k\}_{k=1}^h)$ **as the result obtained by** (15), **we define** $\hat{\mathcal{U}}(\mathcal{A}) = \mathcal{A} \times_1 \hat{U}_1 \times_2 \cdots \times_h \hat{U}_h$. **By the property of subgradient** $\partial\|\cdot\|_{*,\hat{\mathcal{U}}}$ **and the duality between** $\|\mathcal{W}\|_{2,\hat{\mathcal{U}}}$ **and** $\|\mathcal{W}\|_{*,\hat{\mathcal{U}}}$, **we can get the following results.**

**Lemma 4.** *If there exists a dual certificate* $\mathcal{G}$ *(that satisfy* $\Psi_{\mathbb{I}}(\mathcal{G}) = \mathcal{G}$, $P_{\mathbb{T}}(\mathcal{G}) = \mathcal{U}_0 * \mathcal{V}_0^T$ *and* $\|P_{\mathbb{T}^\perp}(\mathcal{G})\|_{2,\hat{\mathcal{U}}} \leq 1$ *), we have*

$$\|\mathcal{M} + \mathcal{H}\|_{*,\hat{\mathcal{U}}} \geq \|\mathcal{M}\|_{*,\hat{\mathcal{U}}} + (1 - \|P_{\mathbb{T}^\perp}(\mathcal{G})\|_{2,\hat{\mathcal{U}}})\|P_{\mathbb{T}^\perp}(\mathcal{H})\|_{*,\hat{\mathcal{U}}}$$

*for any* $\mathcal{H}$ *obeying* $\Psi_{\mathbb{I}}(\mathcal{H}) = 0$.

**Theorem 3.** *If the dual certificate* $\mathcal{G} = \Psi_{\mathbb{I}} P_{\mathbb{T}} (P_{\mathbb{T}} \Psi_{\mathbb{I}} P_{\mathbb{T}})^{-1} (\mathcal{U}_0 * \mathcal{V}_0^T)$ *satisfies* $\|P_{\mathbb{T}^\perp}(\mathcal{G})\|_{2,\hat{\mathcal{U}}} \leq C_1 < 1$ *and* $P_{\mathbb{T}} \Psi_{\mathbb{I}} P_{\mathbb{T}} \succcurlyeq C_2 p \mathcal{I}$, *then we have*

$$\|\mathcal{M} - \hat{\mathcal{X}}\|_F \leq \frac{1}{1 - C_1} \sqrt{\frac{1/C_2 + p}{p} \min(I_1, I_2)} \delta + \delta, \tag{16}$$

*where* $\hat{\mathcal{X}}$ *is obtained by* (15) *and* $p$ *is the sampling rate.*

## 7 COLOR VIDEO INPAINTING

We randomly selected five color video segments with the most rapidly changing frames from category 'run' of the *HMDB51*, including *run* $9^{th}$, *run* $39^{th}$, *run* $40^{th}$, *run* $42^{th}$, and *run* $108^{th}$, and evaluated all tensor completion methods on the selected video segments, where *run* $x^{th}$ is used to represent the $x$-th video in the category 'run'. We present the PSNR values of all methods on the five video segments in Table 1. The results in the table show a significant improvement achieved by our methods (TC-SL and TC-U1) for color video inpainting. The PSNR results obtained by TC-U1 outperform the third-best method (the second-best method is TC-SL) by more than 3.5 dB on average. This substantial improvement showcased by TC-U1 in color video inpainting, as reflected in the higher PSNR values, provides strong evidence for its effectiveness in high-order tensor completion, particularly in scenarios involving non-smooth changes between tensor slices.