# OpenReview forum: "High-Order Tensor Recovery with A Tensor $U_1$ Norm"
_ICLR.cc/2024/Conference — ICLR 2024 Conference Withdrawn Submission_

### Official Review · Reviewer_Yoac · 2023-10-31

**Soundness:** 2 fair
**Presentation:** 2 fair
**Contribution:** 1 poor
**Rating:** 3
**Confidence:** 4

**Summary:**

The paper introduces a tensor recovery technique called Tensor Decomposition Based on Slices-Wise Low-Rank Prior (TDSL) to address challenges in processing high-order tensor data with non-smooth changes. Traditional methods based on t-SVD struggle with non-smooth changes caused by random slice permutation. The proposed TDSL method incorporates a set of unitary matrices to effectively handle permutation variability and non-smoothness. It also introduces a new tensor norm called Tensor $U_1$ norm to extend TDSL to higher-order tensor completion. An optimization algorithm combining the proximal algorithm with the Alternating Direction Method of Multipliers is proposed to solve the resulting models. The convergence of the algorithm is theoretically analyzed. Numerical experiments demonstrate the effectiveness of the proposed method in high-order tensor completion, especially for tensor data with non-smooth changes.

**Strengths:**

**Technical correctness.**  The paper introduces a new tensor recovery technique known as Tensor Decomposition Based on Slices-Wise Low-Rank Prior (TDSL).  The proposed TDSL method demonstrates technical soundness in addressing issues related to permutation variability and non-smoothness in real-world scenarios. It introduces a set of unitary matrices that can be either learned or derived from prior information, facilitating the handling of non-smooth changes resulting from random slice permutation. Additionally, this work introduces a new tensor norm called the Tensor $U_1$ norm, extending TDSL to higher-order tensor completion without requiring additional variables and weights.

**Clear Writing.** The paper maintains a clear and concise writing style, effectively conveying the key concepts and techniques of the proposed method. The use of tables to present experimental results enhances readability and facilitates comparisons of performance across different methods.

**Weaknesses:**

**Limited Novelty.**  While the paper introduces TDSL as a new tensor recovery technique, it's worth noting that similar ideas have already been explored in the literature with clearer motivations and more in-depth theoretical analysis. For instance, reference [R1], authored by Liu G. and Zhang W., investigates the recovery of future data through convolution nuclear norm minimization and provides a comprehensive theoretical foundation.

[R1]. Liu G, Zhang W. Recovery of future data via convolution nuclear norm minimization. IEEE Transactions on Information Theory. 2022 Aug 5;69(1):650-65.

**Limited Theoretical Depth.** The paper does not appear to adequately address the concerns raised regarding its theoretical depth, the derivation of estimation error bounds, and the discussion of sample complexity. These aspects are crucial for a comprehensive understanding of the proposed method and its practical applicability.

**Limited Experimental Evaluations.** The experiments i are limited to a small number of datasets with relatively small sizes. There is a lack of extensive experimentation on large-scale datasets, which can potentially limit the generalizability of the proposed method's performance. Expanding the experimental evaluations to include larger and more diverse datasets would provide stronger empirical support for the approach.

**Limited Significance for ML**. This paper does not provide a clear explanation of how the proposed method can contribute to addressing commonly concerned machine learning issues. As a result, the potential theoretical and empirical significance within the machine learning community may be limited.

**Reproducibility.** While the proposed algorithm contains many details that would be necessary for a reproducible experiment, there is no code provided. This lack of code availability can hinder the ability of other researchers to replicate and verify the results, which is a crucial aspect of scientific reproducibility.

**Questions:**

**Question 1.** Is there a clear explanation of the differences and innovations of the TDSL method compared to previous research, such as the reference [R1]?

**Question 2.** Regarding theoretical depth, can you provide more theoretical background on the TDSL method, especially regarding theoretical explanations related to estimation error bounds and sample complexity?

**Question 3.**  In terms of experimental evaluations, how to expand the experiments to include larger and more diverse datasets to enhance the generalizability of the proposed method's performance?

**Question 4.** Can the authors provide more information on how the TDSL method addresses commonly concerned machine learning issues, emphasizing its importance within the machine learning community?

---

> ### Author Response · Authors · 2023-11-19
>
> Dear Reviewer Yoac,
>
> We appreciate the reviewer’s insightful comment. Here is our detailed reply to your concerns and questions.
>
> **Q1: Is there a clear explanation of the differences and innovations of the TDSL method compared to previous research, such as the reference [R1]?
> [R1]. Liu G, Zhang W. Recovery of future data via convolution nuclear norm minimization. IEEE Transactions on Information Theory. 2022 Aug 5;69(1):650-65.**
>
> **A1**: The proposed method TDSL differs significantly from the approaches outlined in the references provided by the reviewer. First,  ours have the distinct advantage of efficiently utilizing prior data knowledge. The tensor norm given in [R1],  referred to as the convolution nuclear norm, focuses on exploring sparsity priors within tensor data under the Discrete Fourier Transform (DFT), while our proposed TDSL, as defined in (2), investigates slice-wise low-rankness within tensor data. Secondly, the approach in [R1] utilizes the fixed transform  DFT.  In contrast, the transforms employed in our TDSL could be learnable, adhering to unitary constraints, which help to handle the cases involving tensor data exhibiting non-smooth changes.
> $$\boldsymbol{\mathcal{A}}=\boldsymbol{\mathcal{Z_1}}  \times_{k_3} \hat{\boldsymbol{U}}^T_{k_3} \cdots  \times_{k_s} \hat{\boldsymbol{U}}^T_{k_s} \times_{k_{s+1}} \boldsymbol{U}^T_{k_{s+1}} \cdots \times_{k_h} \boldsymbol{U}^T_{k_h}~~~~~~(2)$$
>
> It's noteworthy that the approach in [R1], relying on a fixed transform, encounters a challenge similar to other t-SVD-based tensor recovery methods—namely, performance degradation when dealing with high-order tensor data. This limitation is commonly observed in convolutional methods and t-SVD-based approaches, underscoring the significance of our work.
>
> **Q2: Regarding theoretical depth, can you provide more theoretical background on the TDSL method, especially regarding theoretical explanations related to estimation error bounds and sample complexity?**
>
> **A2**: Following your suggestion,  we have established upper bounds for the estimation error of the two proposed methods in the revised version. The newly incorporated content is presented in Section 4 and the supplementary material. In the subsequent discussion, we provide a brief introduction to the case of TC-SL for given $\boldsymbol{\hat{U}}_{k_n} (n=3,4\cdots s)$. Let us consider a more general case:
>
> $\min_{\boldsymbol{\mathcal{X}},\boldsymbol{U_{k_n}}^T\boldsymbol{U}_{k_n}=\boldsymbol{I}(n=s+1,\cdots,h)}$
>
> $~~~~~~~~~~~~~~~~~~~~~~~~~~~~~~~~~||\boldsymbol{\mathcal{X}} \times_{k_{s+1}} \boldsymbol{U_{k_{s+1}}} \cdots \times_{k_h} \boldsymbol{U_{k_h}}||^{(k_1, k_2)}_{\mathcal{U}, *}$
>
> $~~~~~~~~~~~~~~~~~~~~~~~~~~~~~~~~~s.t. ~ ||\Psi_{\mathbb{I}}(\boldsymbol{\mathcal{M}})-\Psi_{\mathbb{I}}(\boldsymbol{\mathcal{X}})||_F\leq \delta. ~~~~~~~~~~~~~~~~~~~~~~~~~~~~~~~~~~~~~~~~(1)$
>
> Here $\delta \geq 0 $  represents the magnitude of noise present in the tensor data. Then,  we have
> $||\boldsymbol{\mathcal{M}}-\boldsymbol{\hat{\mathcal{X}}}||_F \leq \frac{1}{1-C_1}\sqrt{\frac{1/C_2+p}{p}\min(I_1, I_2)}\delta +\delta$ under conditions in Theorem 3, where $\boldsymbol{\hat{\mathcal{X}}}$ is obtained by (1), $C_1$ and $C_2$ are constant, and $p$ denotes the sampling rate. Therefore, we obtain a theoretical guarantee for the exact recovery of TC-SL by taking $\delta=0$.  (Similar results for TC-U1 can be found in Theorem 2.)
>
>
> **Q3: The experiments i are limited to a small number of datasets with relatively small sizes. There is a lack of extensive experimentation on large-scale datasets, which can potentially limit the generalizability of the proposed method's performance. Expanding the experimental evaluations to include larger and more diverse datasets would provide stronger empirical support for the approach.**
>
> **A3**:  Since this study focuses on addressing the performance degradation of the t-SVD-based methods that are caused by non-smooth cases and to demonstrate our methods can address the issues well, we have used five classical datasets and considered three different application scenes: (1) color images with non-smooth change because of different background, (2) color image sequence disorder, which often happens in the image classification problem,  and (3) color videos with non-smooth change in the frames. Besides, we selected the dataset and determined the amount of experiments in line with the customary practices in conference papers in the field of low-rank recovery (for example, [1-2]). Therefore, we disagree that the experimental evaluations lack diversity or that the datasets are small.
>
> [1] Canyi Lu et. al. Low-rank tensor completion with a new tensor nuclear norm induced by invertible linear transforms.
>
> [2] Jingjing Zheng et. al. Handling slice permutations variability in tensor recovery.

---

> ### Author Response · Authors · 2023-11-19
>
> **Q4: Can the authors provide more information on how the TDSL method addresses commonly concerned machine learning issues, emphasizing its importance within the machine learning community?**
>
> **A4**: This work tackles two common issues seen in t-SVD-based methods, i.e., the performance degradation when dealing with high-order tensor data exhibiting non-smooth changes or imbalance low-rankness. Additionally, it is worth noting that, the proposed tensor decomposition and norms can be applied to many other tensor recovery problems, such as tensor robust PCA, stable tensor completion, and tensor low rank representation. Therefore, we believe that this work significantly contributes to the field of tensor recovery and holds importance for the entire machine learning community. In the revised version, we've clarified this point.
>
> **Q5: This lack of code availability can hinder the ability of other researchers to replicate and verify the results, which is a crucial aspect of scientific reproducibility.**
>
> **A5**: To address your concern, we have made the Matlab code available via the following link:  \url{https://github.com/nobody111222333/MatlabCode.git}.

---

### Official Review · Reviewer_w1M5 · 2023-10-31

**Soundness:** 2 fair
**Presentation:** 2 fair
**Contribution:** 2 fair
**Rating:** 5
**Confidence:** 3

**Summary:**

The paper proposes a new tensor U1 norm based on invertible transforms, and then develops two low-rank tensor completion algorithms based on the transformed tensor and the ADMM, which exploit the slicewise low-rankness. Computational complexity and convergence of the proposed algorithms are discussed. In addition, various numerical experiments on synthetic data test and two real-world applications including image sequence inpainting and color video inpainting have shown the outstanding performance of the proposed algorithms, especially the one based on the tensor U1 norm. The organization of the paper is complete with valid experimental justification.

**Strengths:**

1. The proposed tensor norm has a certain novelty in high order tensorial data completion.
2. Theoretical discussions on the complexity and convergence guarantees may be beneficial for other related works.
3. Real-world applications are important for demonstrating the applicability of the proposed approaches.

**Weaknesses:**

1. Notation clarification should be further improved. For example, the dimensions $I_1,\ldots, I_n$ and the matrix nuclear norm $\|\cdot\|_*$ are not clearly mentioned; and the dimensions of tensors or matrices or the underlying tensor spaces should be clearly presented at least at the beginning to avoid confusion. Minor issues about notation: equation xxx -> (xxx) and some notation could be shortened.
2. The proposed tensor U1 norm seems to be a straightforward extension of dictionary-based L1-norm for vectors/matrices. The connection in this direction was not mentioned or explained.
3. In the abstract, the authors claimed that the proposed methods can handle non-smooth changes. Unfortunately, it cannot be seen explicitly through either the proposed models or the numerical justifications. The motivation of the proposed methods could be explained in more detail. Moreover, a new tensor decomposition was claimed to be introduced in this paper, which does not seem the case since only one slice-wise tensor decomposition form (1) is adopted. Some necessary references for this tensor product are missing.
4. In the numerical experiments, it needs to describe the structure of the underlying data, i.e., whether it's sparse or low-rank in some desired transform.

**Questions:**

1. In the current setting, all the matrices $U_{k_i}$'s are restricted to orthogonal. But can they be extended to unitary matrices, e.g., DFT?
2. In Algorithm 1 line 3, where do the matrices $U$ and $V$ come from? They are never used in the other lines of the entire algorithm.
3. The tensor norm in (3) is based on the across-slice wise low-rankness under certain transform, i.e., transformed low-rankness, but the tensor U1 norm in (9) is to describe the transformed sparsity. What are the connections between them? If the data set is only sparse or only low-rank, then would it be unfair to compare both in the same context? Furthermore, why is TC-U1 always performing better than the TC-SL? Is that due to the low-rankness nature for the testing data sets instead of sparsity?

---

> ### Author Response · Authors · 2023-11-19
>
> Dear Reviewer w1M5,
>
> We appreciate the reviewer’s insightful comment. Here is our detailed reply to your concerns and questions.
>
> **Q1: What are the connections between the two proposed methods?**
>
> **A1**:   For your question, we explain as follows.
> $$\boldsymbol{\mathcal{A}}=\boldsymbol{\mathcal{Z_1}}  \times_{k_3} \hat{\boldsymbol{U}}^T_{k_3} \cdots  \times_{k_s} \hat{\boldsymbol{U}}^T_{k_s} \times_{k_{s+1}} \boldsymbol{U}^T_{k_{s+1}} \cdots \times_{k_h} \boldsymbol{U}^T_{k_h}~~~~~~(2)$$
>
> From the proposed TDSL given in (2), we observe that TDSL of $\boldsymbol{\mathcal{A}}$ depends on the choice of  $k_1$ and $k_2$, and it considers different kinds of low-rankness of information in the tensor data for different $(k_1, k_2)$. If we only consider one mode, it may lead to the loss of correlation information across the remaining modes. On the other hand, inspired by the sparsity prior in natural signals, i.e., the natural signals are often sparse if expressed on a proper basis,   we propose an alternative approach by assuming that $\boldsymbol{\mathcal{Z_1}}$ exhibits sparsity along its $(k_1, k_2)$-th mode, i.e., there exists $\boldsymbol{U_{k_1}}$ and $\boldsymbol{U_{k_2}}$ such that  $\boldsymbol{\mathcal{Z_2}}= \boldsymbol{\mathcal{Z_1}} \times_{k_1} \boldsymbol{U_{k_1}} \times_{k_2} \boldsymbol{U_{k_2}}$ is sparse.  Thus, we obtained TDST given in (5).
>
>    $$  \boldsymbol{\mathcal{A}}=\boldsymbol{\mathcal{Z_2}} \times_{k_1}  \boldsymbol{U}^T_{k_1}  \times_{k_2} \boldsymbol{U}^T_{k_2} \times_{k_3} \hat{\boldsymbol{U}}^T_{k_3} \cdots  \times_{k_s} \hat{\boldsymbol{U}}^T_{k_s} \times_{k_{s+1}} \boldsymbol{U}^T_{k_{s+1}} \cdots \times_{k_h} \boldsymbol{U}^T_{k_h}. ~~~~~~~~~~~~~~~~~~(5)$$
>
> **Q2: The connection in the direction of dictionary-based L1-norm for vectors/matrices was not mentioned or explained.**
>
> **A2**: In introducing the proposed TDST, we utilized the sparsity prior observed in natural signals, where such signals tend to exhibit sparsity when expressed on an appropriate basis. In the revised manuscript, we have included a more detailed explanation.
>
> **Q3: The authors claimed that the proposed methods can handle non-smooth changes. Unfortunately, it cannot be seen explicitly through either the proposed models or the numerical justifications. The motivation of the proposed methods could be explained in more detail.**
>
> **A3**: We have given the discussion in Sections 2.1-2.2 to introduce the proposed methods from the angle of handling the non-smooth changes and studying low rankness across different dimensions of tensor data. To demonstrate this point, we have used five classical datasets and considered three different application scenes: (1) color images with non-smooth change because of different backgrounds, (2) color image sequence disorder, which often happens in the image classification problem, and (3) color videos with non-smooth change in the frames. The experimental results demonstrate the effectiveness of the proposed methods.
>
> **Q4: a new tensor decomposition was claimed to be introduced in this paper, which does not seem the case since only one slice-wise tensor decomposition form (1) is adopted. Some necessary references for this tensor product are missing.**
>
> **A4**: The proposed new tensor decomposition methods, including TDSL and TDST, have been given in (2) and (5) in the original version, respectively. In response to your comment, we have included references for the tensor product.  Further detailed discussions regarding tensor product-based methods have been included in the supplementary material due to space constraints.
>
> **Q5: In the numerical experiments, it needs to describe the structure of the underlying data, i.e., whether it's sparse or low-rank in some desired transform.**
>
> **A5**: In our real application experiments, we did not have a preference for whether the data was inherently sparse or low-rank.  Instead, we selected high-order tensor data with non-smooth change due to the objective of this work.  For example, the BSD dataset was chosen by following [1], and it has non-continuous changes because of varying backgrounds.
>
> [1] Canyi et.al. Tensor robust principal component analysis with a new tensor nuclear norm. IEEE Transactions on Pattern Analysis and Machine Intelligence.
>
> **Q6: If the data set is only sparse or only low-rank, then would it be unfair to compare both in the same context? Furthermore, why is TC-U1 always performing better than the TC-SL?**
>
> **A6**: As previously explained in response to Q5, in our real application experiments, we did not have a preference for whether the data was inherently sparse or low-rank. TC-U1 outperforms TC-SL in most cases because TC-SL only considers low-rankness along one dimension of the tensor data, potentially leading to the loss of correlation information across its remaining dimensions.

---

> > ### Author Response · Authors · 2023-11-19
> >
> > **Q7: In the current setting, all the matrices 's are restricted to orthogonal. But can they be extended to unitary matrices, e.g., DFT?**
> >
> > **A7**: In this study, we have considered unitary matrices and employed the Discrete Fourier Transform (DFT) in our experiments.
> >
> > **Q8: In Algorithm 1 line 3, where do the matrices and come from?**
> >
> > **A8**: The matrices $\boldsymbol{U}$ and $\boldsymbol{V}$ can be obtained by SVD of
> >
> > the product of $[\sqrt{\mu^{(t)}}\boldsymbol{\mathcal{A_{({k_n})}}}, \sqrt{\eta^{(t)}}\boldsymbol{U^{(t)}_{k_n}}]$
> >
> > and $[\sqrt{\mu^{(t)}}\boldsymbol{\mathcal{B}}_{({k_n})}, \sqrt{\eta^{(t)}}\boldsymbol{I}]^T$,
> >
> > where $\boldsymbol{\mathcal{A}}=\boldsymbol{\mathcal{Z^{(t+1)}}}\times_{k_h}\cdots \times_{k_n+1}\\boldsymbol{U^{(t)T}_{k_n+1}}$
> >
> > and
> >
> > $\boldsymbol{\mathcal{B}}=\boldsymbol{\hat{\mathcal{P}}}\times_{s+1}\cdots \times_{{k_n}-1} \boldsymbol{U}^{(t+1)}_{{k_n}-1}$. It has been introduced in the detailed introduction of the Step 2 of the algorithm.
> >
> > **Q9: Notation clarification should be further improved.**
> >
> > **A9**:  Following your suggestion, we have clarified the notations and meticulously reviewed the manuscript to avoid any ambiguities.

---

### Official Review · Reviewer_1gNF · 2023-11-01

**Soundness:** 3 good
**Presentation:** 2 fair
**Contribution:** 2 fair
**Rating:** 5
**Confidence:** 4

**Summary:**

This paper studies the t-SVD based low-rank tensor recovery problem under the potential non-smooth changes along the third dimension. To tackle that, the authors introduce a new tensor $U_1$ norm with a set of {$U_{k_n}$} given by the prior slides-wise low-rank information. Some, but not comprehensive, theoretical results have been established, together with a good load of numerical experiments.

**Strengths:**

The paper aims at an interesting new angle of t-SVD based tensor recovery problem. As far as I read, the analysis is overall correct.

**Weaknesses:**

Although the analysis is about right, the results rely on a set of accurate slice-wise low-rank prior. That leads to major questions:

1. How to efficiently obtain those priors? Please give examples. Based on what I read in the numerical experiment section. It seems the authors don't have a practical way to obtain the required prior {$U_{k_n}$}.

2. The analysis doesn't show what could happen if the inaccurate low-rank prior was used. There is a chance of having even worse recovery results when some large noise is added to the prior.

Therefore, I doubt the proposed $U_1$ norm is practically useful at all.

The paper was not carefully written, many notations were used before being defined. For example,

3. Page 3 under (1), $(I_{k_1}, I_{k_2})$ were not defined until page 4 Definition 1.

4. Page 3 above (2), the full name of DCM were not introduced until page 7 Sec 4.1.

5. Page 4 'By known result in Luet al. (2019b)', although I am familiar with the paper, the author should restate the quoted result for the completeness of the analysis.

6. Page 7, 'For given $R$' -> 'For given rank parameter $R$'

**Questions:**

1. The authors tried to motivate the 'non-smooth changes' problem with a 'random slice permutations' story. It is not clear to why random slice permutations may happen in real applications. Please be more specific about why the image classification task will cause random slice permutations.

---

> ### Author Response · Authors · 2023-11-19
>
> Dear Reviewer 1gNF,
>
> We appreciate the reviewer’s insightful comment. Here is our detailed reply to your concerns and questions.
>
> **Q1: How to efficiently obtain those priors? Please give examples. Based on what I read in the numerical experiment section. It seems the authors don't have a practical way to obtain the required prior  $\{\boldsymbol{U_{k_n}}\}$**
>
> **A1**: The assumptions regarding smoothness priors used in this work are really straightforward.  For example, in real-world applications of our experiments,  the parameters in   TC-U1 were configured as $\\{s=2, (\boldsymbol{\hat{U_{k_1}}},\boldsymbol{\hat{U_{k_2}}})=(\boldsymbol{F_{I_1}},\boldsymbol{F_{I_2}})\\}$. This configuration is based on the assumption that the first two dimensions of images exhibit smoothness characteristics.  In the revision, we have clarified this point to avoid any confusion and ensure clarity.
>
> **Q2: The analysis doesn't show what could happen if the inaccurate low-rank prior was used. There is a chance of having even worse recovery results when some large noise is added to the prior.**
>
> **A2**:  To address your concern,  in our revised manuscript, we have proposed our analysis to consider scenarios where the tensor data may be contaminated by noise or inaccurately represented as low-rank.
> We introduce Stable TC-SL (STC-SL) and Stable TC-U1 (STC-U1) and establish upper bounds of estimation error by  the two proposed methods. These additions are detailed in Section 4 and the supplementary material. As a brief overview, we discuss the case of STC-U1 based on given  $\boldsymbol{\hat{U}_{k_n}} (n=3,4,\cdots,s)$, which is formulated as
>
> $\min_{\boldsymbol{\mathcal{X}},\boldsymbol{U_{k_n}}^T\boldsymbol{U}_{k_n}=\boldsymbol{I}(n=s+1,\cdots,h)}$
>
> $~~~~~~~~~~~~~~~~~~~~~~~~~~~~~~~~~||\boldsymbol{\mathcal{X}} \times_{k_{s+1}} \boldsymbol{U_{k_{s+1}}} \cdots \times_{k_h} \boldsymbol{U_{k_h}}||_{\mathcal{U}, 1}$
>
> $~~~~~~~~~~~~~~~~~~~~~~~~~~~~~~~~~s.t. ~ ||\Psi_{\mathbb{I}}(\boldsymbol{\mathcal{M}})-\Psi_{\mathbb{I}}(\boldsymbol{\mathcal{X}})||_F\leq \delta. ~~~~~~~~~~~~~~~~~~~~~~~~~~~~~~~~~~~~~~~~(1)$
>
> Here $\delta \geq 0 $  represents the magnitude of the noise. Under the conditions of Theorem 2, we can deduce
> $||\boldsymbol{\mathcal{M}}-\boldsymbol{\hat{\mathcal{X}}}||_F \leq \frac{1}{1-C_1}\sqrt{\frac{1/C_2+p}{p}I_1I_2}\delta +\delta$, where    $\boldsymbol{\hat{\mathcal{X}}}$ is obtained by (1), $C_1$ and $C_2$ are constant, and
>   $p$ denotes the sampling rate.
>
> From the case of STC-U1, we can see that, although we only take tensor completion as example in our original submission,  the proposed tensor norms can be directly applied to many other tensor recovery problems (such as tensor robust PCA, stable tensor completion, and tensor low-rank representation) for handling many situations, like noiseless, sparse noise, Gaussian noise.   The above case is about stable tensor completion.  In the revision, we have clarified this point.
>
>  The STC-U1 case illustrates that our proposed tensor norms are not only applicable to tensor completion with no noise,   but also can extend to a variety of other tensor recovery problems, such as tensor robust PCA, stable tensor completion, and tensor low-rank representation. These methods are effective in various scenarios and in handling different types of noise.
>
> **Q3: The authors tried to motivate the 'non-smooth changes' problem with a 'random slice permutations' story. It is not clear to why random slice permutations may happen in real applications. Please be more specific about why the image classification task will cause random slice permutations.**
>
> **A3**: Let $\boldsymbol{\mathcal{M}} \in \mathbb{R}^{I_1 \times I_2 \times I_3 \times I_4}$ be a tensor data constructed by the classification data, where  $I_1 \times I_2$  represents the size of each image sample, $I_3 = 3$ is RGB channel number
> of each image, and $I_4$ is the number of image samples. The scenario of random slice permutations is frequently observed within classification tasks, as the sequence of samples is often disordered prior to classification. In the revision, we have clarified this point to avoid any confusion.
>
>
> **Q4: The paper was not carefully written, many notations were used before being defined.**
>
> **A4**: We sincerely appreciate the reviewer's meticulous proofreading. For all unclear notations pointed out by the reviewer, we have provided the corresponding explanations in the revised version. Additionally, we have thoroughly checked the manuscript to avoid any unclear notations.